# Understanding Constraint Inference in Safety-Critical Inverse Reinforcement Learning

**Bo Yue[1], Shufan Wang[2], Ashish Gaurav[3,4], Jian Li[2], Pascal Poupart[3,4], Guiliang Liu[1]**[*]
[1]School of Data Science, The Chinese University of Hong Kong, Shenzhen,
[2]Stony Brook University, [3]University of Waterloo, [4]Vector Institute
`boyue@link.cuhk.edu.cn, liuguiliang@cuhk.edu.cn,`
`{shufan.wang,jian.li.3}@stonybrook.edu,`
`{a5gaurav,ppoupart}@uwaterloo.ca`

## ABSTRACT

In practical applications, the underlying constraint knowledge is often unknown and difficult to specify. To address this issue, recent advances in Inverse Constrained Reinforcement Learning (ICRL) have focused on inferring these constraints from expert demonstrations. However, the ICRL approach typically characterizes constraint learning as a tri-level optimization problem, which is inherently complex due to its interdependent variables and multiple layers of optimization. Considering these challenges, a critical question arises: *Can we implicitly embed constraint signals into reward functions and effectively solve this problem using a classic reward inference algorithm?* The resulting method, known as Inverse Reward Correction (IRC), merits investigation. In this work, we conduct a theoretical analysis comparing the sample complexities of both solvers. Our findings confirm that the IRC solver achieves lower sample complexity than its ICRL counterpart. Nevertheless, this reduction in complexity comes at the expense of generalizability. Specifically, in the target environment, the reward correction terms may fail to guarantee the safety of the resulting policy, whereas this issue can be effectively mitigated by transferring the cost functions via the ICRL solver. Advancing our inquiry, we investigate conditions under which the ICRL solver ensures $\varepsilon$-optimality when transferring to new environments. Empirical results across various environments validate our theoretical findings, underscoring the nuanced trade-offs between complexity reduction and generalizability in safety-critical applications.

## 1 INTRODUCTION

Aligning the decision process with the underlying constraints in the environment is a crucial prerequisite for solving decision-making problems in safety-critical applications. To realize this vision, safe Reinforcement Learning (RL) algorithms typically optimize a control policy based on a known or manually-specified constraint (Liu et al., 2021; Gu et al., 2022). However, the ground truth constraints are often unknown in many real-world applications. Moreover, given the inherent complexity of environmental dynamics, safety constraints must accurately model the interdependencies among numerous variables, which are difficult to capture solely with prior knowledge.

To resolve the above challenges, Inverse Constrained Reinforcement Learning (ICRL) designs a data-driven constraint inference method to learn the constraints from expert demonstrations (Scobee & Sastry, 2020). Specifically, an ICRL algorithm (Malik et al., 2021) typically addresses a tri-level optimization problem involving the update of 1) the feasibility functions to represent constraints, 2) the Lagrange parameters to balance reward maximization and constraint satisfaction, and 3) the policy function to guide the agent's behaviors. Under this setting, the variables subject to optimization are interdependent, and the sub-optimality of one variable can influence the performance of the others. To mitigate the complexity of this problem, Hugessen et al. (2024) proposed to simplify the ICRL solver by incorporating the impact of both constraint and Lagrange parameters into a reward correction term. This modification reduces the ICRL solver to a bi-level solver known as Inverse Reward Correction (IRC) (Li et al., 2023). Hugessen et al. (2024) empirically demonstrated that

---

[*]Corresponding author: Guiliang Liu

such simplification does not compromise the performance of constraint learning. These observations raise significant questions about the necessity of explicitly modeling constraints. It is still uncertain whether the canonical reward learning framework is sufficient to capture an agent's preferences within a Constrained Markov Decision Process (CMDP).

To address this issue, in this work, we conduct a rigorous study to understand the impact of modeling constraints. Specifically, we theoretically and empirically compare the performance of IRC and ICRL from the following perspectives:

**Training Efficiency.** Unlike previous studies, which primarily compared IRC and ICRL through empirical evaluations (Hugessen et al., 2024), we are the first to provide a theoretical quantification of training efficiency for both IRC and ICRL solvers in capturing the safety preferences in expert agents' decisions. However, achieving this objective is inherently challenging due to the unidentifiability of the optimal solution, as multiple solutions can explain expert demonstrations (Ng et al., 2000). To address this, rather than modeling pointwise solutions, we adopt the approach first proposed by Metelli et al. (2021), which introduces a theoretical framework that characterizes a complete set of feasible solutions. This approach enables us to determine the intrinsic sample complexity of both IRC and ICRL without being obfuscated by further restrictions to ensure the uniqueness of optimal solutions. Our results indicate that ICRL requires more training samples, consistent with the empirical findings in Hugessen et al. (2024). Moreover, our theoretical analysis provides a deeper understanding of the increased complexity. Specifically, we show that ICRL solvers capture constraint-violating movements by leveraging known reward signals. During the update of a constrained RL policy, these movements correspond to decision-making patterns that drive an increase in the Lagrange multipliers.

**Cross-environment Transferability.** While previous constraint learning methods primarily focus on imitating experts' behavior (Scobee & Sastry, 2020; Liu et al., 2023; Hugessen et al., 2024; Liu et al., 2024), a key motivation for inferring experts' safety preferences is to generalize this knowledge for guiding policy learning in similar contexts (Feng et al., 2023; Zhang et al., 2024). In this study, we compare IRC and ICRL in terms of guaranteeing *safety* and *optimality* of policies based on inferred terms across different environments. We begin with an illustrative example to demonstrate situations where IRC could potentially induce unsafe behaviors. We identify and summarize the conditions under which IRC leads to such unsafe behaviors and explain why the performance of ICRL is more robust by explicitly modeling constraints. Regarding the optimality of generalized policies, we theoretically investigate how mismatches in environmental dynamics and reward signals influence the acquisition of optimal policies based on learned cost functions by the ICRL solver. In the absence of constraint satisfaction, these results cannot be extended to IRC.

**Contributions.** We compare the training efficiency and cross-environment transferability of IRC and ICRL solvers to address the critical question: *Can we implicitly embed constraint signals into reward functions and effectively solve this problem using a classic reward inference algorithm?*

- Following the theoretical framework from Metelli et al. (2021), we introduce the IRC solver (Definition 3.3) to overcome the limitation of the IRL solver, which lacks a mechanism to leverage existing reward signals and is not compatible with different rewards. Furthermore, we analyze the sample complexity of IRC (Section 4.1) and compare it with existing results for ICRL (Yue et al., 2024), showing that IRC achieves lower sample complexity than ICRL under the same optimality criterion (Theorem 4.2).

- We conduct a formal study of transferability in safety to compare IRC and ICRL. We show that transferred cost functions by ICRL are guaranteed to preserve safety in overlapping critical regions (Lemma 5.2), whereas transferred reward correction terms by IRC can be offset by the difference in reward functions and transition dynamics between source and target environments (Theorem 5.3).

- We analyze the optimality of transferred cost functions in target environments, by extending the transferability definition from Schlaginhaufen & Kamgarpour (2024, Definition 3.1) in regular MDP settings to CMDP settings. Specifically, we define the suboptimality gap under CMDP settings (Definition 5.5). Building on this, we derive a sufficient condition that limits the similarity between source and target environments to guarantee $\varepsilon$-optimality for ICRL (Theorem 5.7).

- Finally, we empirically validate our results on training efficiency and cross-environment transferability in various environments (Section 6).

## 2    RELATED WORK

**Inverse Constrained Reinforcement Learning (ICRL).** Unlike IRL, which focuses primarily on recovering reward functions, ICRL seeks to align with expert agents' preferences by inferring the constraints they adhere to. The majority of existing ICRL algorithms update cost functions by maximizing the likelihood of generating expert demonstrations under the maximum (casual) entropy framework (Scobee & Sastry, 2020). Subsequent works have extended this approach from discrete to continuous state-action spaces (Malik et al., 2021; Qiao et al., 2023; Xu & Liu, 2024b; Quan et al., 2024; Zhao et al., 2025). To enhance training efficiency, Liu & Zhu (2022); Gaurav et al. (2023) combined ICRL with bi-level optimization techniques. Towards theoretical groundings of ICRL, Yue et al. (2024) recently proposed efficient constraint inference through exploration strategies, achieving tractable sample complexity. However, these works primarily evaluate their performance by applying inferred constraints within the same environment used for learning. While Xu & Liu (2024a) proposed an approach to learning a robustly safe policy across a set of pre-defined and limitedly varied transition dynamics, the challenge of transferring constraints to new environments remains largely unexplored.

**Transferability in Inverse Reinforcement Learning (IRL).** A significant application of IRL algorithms is to guide policy learning in similar environments. However, obtaining guarantees of transferability in unregularized settings is more challenging than in entropy-regularized contexts. To facilitate knowledge transfer in unregularied settings, Metelli et al. (2021) assumed that the feasible reward functions recovered remain valid in target environments, and Amin et al. (2017) reduced the dimension of the reward class to state-only rewards. In contrast, entropy-regularized settings have been explored more extensively. Cao et al. (2021) and Skalse et al. (2023) demonstrated that under entropy regularization, the expert's reward can be identified up to potential shaping transformations (Ng et al., 1999). In addition, Rolland et al. (2022) showed that to guarantee transferability across any transition laws, the expert's reward must be identified up to a constant. Building upon these insights, Cao et al. (2021) and Rolland et al. (2022) learned the reward function from multiple experts who shared rewards but had sufficiently different transition laws distinguished by a specific rank condition. Continuing this line of research, Schlaginhaufen & Kamgarpour (2024) extended this approach to an offline setting and derived a sufficient condition for transferability considering local changes in the transition law when learning from a single expert. However, these methods focus on transferring reward functions in regular MDPs, without addressing the generalization of constraints in CMDPs.

## 3    PRELIMINARIES AND PROBLEM FORMULATION

**Notation.** Let $\mathcal{X}$ and $\mathcal{Y}$ be two sets. The notation $\mathcal{Y}^{\mathcal{X}}$ represents the set of functions $f : \mathcal{X} \to \mathcal{Y}$. The set of probability measures over $\mathcal{X}$ is denoted as $\Delta^{\mathcal{X}} = \{\nu \in [0,1]^{\mathcal{X}} : \sum_{x \in \mathcal{X}} \nu(x) = 1\}$ and we denote $\Delta_{\mathcal{Y}}^{\mathcal{X}}$ as the set of functions: $\mathcal{Y} \to \Delta^{\mathcal{X}}$. We define $\min_{x \in \mathcal{X}}^{+} f(x)$ returns the minimum positive value of $f$ over $\mathcal{X}$. For a linear operator $T$, its image is denoted by $\mathrm{im}(T)$. Let $c^E$ represent the ground-truth cost function obeyed by the expert, and $\mu^E$ denote the expert occupancy measure. Let $\mathbb{1}(\cdot)$ denote the indicator function. The expansion operator $E : \mathbb{R}^{\mathcal{S}} \to \mathbb{R}^{\mathcal{S} \times \mathcal{A}}$ satisfies $(Ef)(s, a) = f(s)$. The complete notation is provided in Appendix Table 1.

**Constrained Markov Decision Process (CMDP).** The environment is modeled as a stationary CMDP $\mathcal{M}_c := (\mathcal{S}, \mathcal{A}, P_{\mathcal{T}}, r, c)$, where $\mathcal{S}$ and $\mathcal{A}$ are the finite state and action spaces, with cardinalities $S = |\mathcal{S}|$ and $A = |\mathcal{A}|$; $P_{\mathcal{T}}(s'|s, a) \in \Delta_{\mathcal{S} \times \mathcal{A}}^{\mathcal{S}}$ defines the transition dynamic; $r \in [0, R_{\max}]^{\mathcal{S} \times \mathcal{A}}$ and $c \in [0, C_{\max}]^{\mathcal{S} \times \mathcal{A}}$ denote the reward and cost functions. The set $\{\epsilon, \mu_0, \gamma\}$ represents additional environmental configurations, where $\epsilon$ defines the threshold of the constraint, $\mu_0 \in \Delta^{\mathcal{S}}$ denotes the initial state distribution, and $\gamma \in [0, 1)$ is the discount factor. The agent's policy $\pi \in \Delta_{\mathcal{S}}^{\mathcal{A}}$. We focus on the infinity planning horizon, and our theoretical results are based on a discrete state-action space.

**Value and advantage functions.** We denote action value functions in a CMDP $\mathcal{M}_c$ for costs and rewards as $Q_{\mathcal{M}_c}^{c, \pi}$ and $Q_{\mathcal{M}_c}^{r, \pi}$. The superscript $c$ or $r$ specifies the actual costs or rewards evaluated. The reward action-value function $Q_{\mathcal{M}_c}^{r, \pi}(s, a) = \mathbb{E}_{\pi, P_{\mathcal{T}}}\left[\sum_{t=0}^{\infty} \gamma^t r(s_t, a_t) | s_0 = s, a_0 = a\right]$, and the reward advantage function $A_{\mathcal{M}_c}^{r, \pi}(s, a) = Q_{\mathcal{M}_c}^{r, \pi}(s, a) - V_{\mathcal{M}_c}^{r, \pi}(s)$, where the reward state-value function $V_{\mathcal{M}_c}^{r, \pi}(s) = \mathbb{E}_{\pi}[Q_{\mathcal{M}_c}^{r, \pi}(s, a)]$. The same notation scheme applies to cost value functions in $\mathcal{M}_c$ by

replacing the superscript $r$ with $c$, i.e., $Q_{\mathcal{M}_c}^{c,\pi}$ and $V_{\mathcal{M}_c}^{c,\pi}$. This scheme also applies to CMDP without knowing the cost, i.e., $\mathcal{M} = \mathcal{M}_c \backslash c$ by replacing the subscript $\mathcal{M}_c$ with $\mathcal{M}$, i.e., $Q_{\mathcal{M}}^{r,\pi}$, $V_{\mathcal{M}}^{r,\pi}$ and $A_{\mathcal{M}}^{r,\pi}$.

**Constrained Reinforcement Learning (CRL).** Within a CMDP environment, CRL learns a policy $\pi$ that maximizes the cumulative rewards subject to a known constraint:

$$\mathsf{CRL}(r, c) = \max_{\pi} \; \mathbb{E}_{\mu_0, \pi, P_{\mathcal{T}}} \Big[ \sum_{t=0}^{\infty} \gamma^t r(s_t, a_t) \Big] \qquad \text{s.t.} \;\; \mathbb{E}_{\mu_0, \pi, P_{\mathcal{T}}} \Big[ \sum_{t=0}^{\infty} \gamma^t c(s_t, a_t) \Big] \le \epsilon, \qquad \text{(PI)}$$

where $\epsilon = 0$ indicates a hard constraint and $\epsilon > 0$ represents a soft constraint.

**Inverse Constraint Inference (ICI).** In many practical applications, constraints are not readily available, so we need to infer the constraints followed by expert agents based on their behaviors. An ICI problem is a pair $\mathfrak{P} = (\mathcal{M}, \pi^E)$ where $\pi^E \in \Delta_{\mathcal{S}}^{\mathcal{A}}$ is the expert's policy.

A common solver for this ICI problem under the RL setting is known as Inverse Constrained Reinforcement Learning (ICRL), which is formally defined as follows:

**Definition 3.1.** *(ICRL solver for ICI ([Malik et al., 2021](#)))*. An ICRL solver is denoted as $\mathbb{S}_{\text{ICRL}}$. A cost function $c$ is a *feasible* solution for an ICI problem $\mathfrak{P}$ if and only if $\pi^E$ is an optimal policy for CMDP $\mathcal{M}_c$. We denote by $\mathcal{C}_{\mathfrak{P}}$ the set of feasible cost functions derived by $\mathbb{S}_{\text{ICRL}}(\mathfrak{P})$.

Previous ICRL solvers ([Scobee & Sastry](#), 2020; [Malik et al., 2021](#)) explicitly model constraints and infer the cost function by alternatively optimizing the policy and the cost function. In the phase of policy optimization, they commonly solve a CRL problem ([PI](#)) by studying its Lagrangian dual:

$$D\left[\mathsf{CRL}(r, c)\right] = \min_{\lambda > 0} \max_{\pi} \mathcal{J}(\pi, r - \lambda c) + \lambda \epsilon, \qquad \text{(DI)}$$

where $\mathcal{J}(\pi, r - \lambda c) = \mathbb{E}_{\mu_0, \pi, P_{\mathcal{T}}} \Big[ \sum_{t=0}^{\infty} \gamma^t \Big( r(s_t, a_t) - \lambda c(s_t, a_t) \Big) \Big]$.

**Theorem 3.2.** *(CRL has zero duality gap ([Paternain et al., 2019](#)))*. *Suppose that $r$ and $c$ are bounded and the Slater's condition holds for ([PI](#)), then strong duality holds for ([PI](#)), i.e., $PI^* = DI^*$.*

Accordingly, the optimal policy in CRL objective ([PI](#)) can be equivalently solved by utilizing an *unconstrained* objective ([DI](#)). Based on the dual representation of CRL problem, ICRL solvers are essentially solving a tri-level optimization problem ([Kim et al., 2023](#)):

$$\max_{c} \max_{\lambda} \min_{\pi} \mathcal{J}(\pi^E, r - \lambda c) - \mathcal{J}(\pi, r - \lambda c). \qquad (1)$$

Given the complexity of tri-level optimization, [Hugessen et al. (2024)](#) recently explored whether we can 1) apply an IRL algorithm to recover $\tilde{r} = r - \lambda c = r - \tilde{c}$ by optimizing $\lambda$ and $c$ collectively in $\tilde{c} = \lambda c$ if the range of $\tilde{c}$ is a convex cone, and 2) learn an imitation policy by directly maximizing the cumulative rewards $\mathbb{E}[\sum_{t=0}^{\infty} \gamma^t \tilde{r}(s_t, a_t)]$ without considering the constrained optimization objective. In this work, we formally define this method as Inverse Reward Correction (IRC) as follows:

**Definition 3.3.** *(IRC solver for ICI ([Li et al., 2023](#)))*. An IRC solver is denoted as $\mathbb{S}_{\text{IRC}}$. A reward correction term $\Delta r$ is a *feasible* solution for an ICI problem $\mathfrak{P}$ if and only if $\pi^E$ is an optimal policy for $(\mathcal{M} \backslash r) \cup r^c$, where corrected rewards $r^c(s, a) = r(s, a) + \Delta r(s, a), \forall (s, a)$. We denote by $\mathcal{R}_{\mathfrak{P}}$ the set of feasible reward correction terms derived by $\mathbb{S}_{\text{IRC}}(\mathfrak{P})$.

For the sake of clarity, we simplify $(\mathcal{M} \backslash r) \cup r^c$ to $\mathcal{M} \cup r^c$ in the remainder of the paper. Under this setting, the correction term can play the role of negative collective cost function such that $\Delta r = -\tilde{c}$. If the negative optimal $\tilde{c}$ can be represented within the bounded range of the correction term, i.e., $-\tilde{c}^* = -\lambda^* c^* \in \text{range}(\Delta r)$, where $\lambda^*$ and $c^*$ are optimal solutions of ([1](#)), the tri-level optimization can be transferred to a bi-level one as defined in the following:

$$\max_{\Delta r} \min_{\pi} \mathcal{J}(\pi^E, r + \Delta r) - \mathcal{J}(\pi, r + \Delta r). \qquad (2)$$

[Hugessen et al. (2024)](#) demonstrated that this simplification results in a more performant solver. In the following sections, we will provide a more formal comparison of these solvers, focusing on their *sample complexity* and *transferability*, i.e., performance in transferring to new environments.

## 4 Training Efficiency: A Formal Study of Sample Complexity

In this section, we compare the training efficiency of ICRL and IRC solvers by deriving their sample complexity and analyzing their performance gaps.

## 4.1 Sample Complexity of IRC Solver

To compute the sample complexity of the IRC solver, we adopt the theoretical framework from Metelli et al. (2021) and define the feasible set of reward correction terms as follows:

**Lemma 4.1.** *(Feasible reward correction set implicit). Let $\mathfrak{P}$ be an ICI problem and $\mathbb{S}_{IRC}$ be the IRC solver. $\Delta r$ is a feasible reward correction term, i.e., $\Delta r \in \mathcal{R}_{\mathfrak{P}}$ if and only if $\forall (s,a) \in \mathcal{S} \times \mathcal{A}$:*

*(i) if $\pi^E(a|s) > 0$, then $Q^{r+\Delta r, \pi^E}_{\mathcal{M} \cup (r+\Delta r)}(s,a) = V^{r+\Delta r, \pi^E}_{\mathcal{M} \cup (r+\Delta r)}(s)$,*

*(ii) if $\pi^E(a|s) = 0$, then $Q^{r+\Delta r, \pi^E}_{\mathcal{M} \cup (r+\Delta r)}(s,a) \leq V^{r+\Delta r, \pi^E}_{\mathcal{M} \cup (r+\Delta r)}(s)$.*

Intuitively, in a CMDP, reward function $r$ alone does not align with the expert policy $\pi^E$ due to potential constraint violations, i.e. $\pi^E(a|s) = 0$ but $Q^{r, \pi^E}_{\mathcal{M}}(s,a) > V^{r, \pi^E}_{\mathcal{M}}(s)$. The correction term $\Delta r$ adjusts the reward $r$ so that $r + \Delta r$ collectively ensures the optimality of the expert policy. This approach differs from the IRL solver (Metelli et al., 2021; Lindner et al., 2022) in two key aspects: 1) IRL lacks a mechanism to leverage the known reward function $r$ for constraint inference, and 2) IRL is not compatible with different reward signals in new environments.

To provide a fair comparison of sample complexity between the two solvers, we study a *uniform sampling strategy*, detailed in Appendix Algorithm 1. This strategy queries the generative model to sample the state-action space, enabling the estimation of the transition dynamics and the expert policy as $\widehat{\mathfrak{P}} = (\widehat{\mathcal{M}}, \widehat{\pi}^E)$, where $\widehat{\mathcal{M}} = (\mathcal{M} \backslash P_{\mathcal{T}}) \cup \widehat{P_{\mathcal{T}}}$. Let $\delta \in (0,1)$ be the significance level. This strategy guarantees that with probability greater than $1 - \delta$, the Hausdorff distance $d_H$ between the ground truth and estimated feasible set is upper bounded by an amount related to the number of samples.

In the case of the IRC solver, this Hausdorff distance between $\mathcal{R}_{\mathfrak{P}}$ and $\mathcal{R}_{\widehat{\mathfrak{P}}}$ is upper bounded by

$$d_H(\mathcal{R}_{\mathfrak{P}}, \mathcal{R}_{\widehat{\mathfrak{P}}}) \leq \max_{(s,a) \in \mathcal{S} \times \mathcal{A}} \mathcal{I}^{\Delta r}_{k+1}(s,a), \text{ with } \mathcal{I}^{\Delta r}_{k+1}(s,a) = \frac{2\gamma R_{\max}}{1 - \gamma} \sqrt{\frac{2 \ell_{k+1}(s,a)}{N^+_{k+1}(s,a)}}, \quad (3)$$

where $N^+_{k+1}(s,a)$ is the positive cumulative count of visitations to $(s,a)$ (formally defined in Appendix B.2) and $\ell_{k+1}(s,a) = \log \left( 12 S A (N^+_{k+1}(s,a))^2 / \delta \right)$. Towards reducing this upper bound below a targeted accuracy, we derive the sample complexity of the IRC solver.

**Theorem 4.2.** *(Sample Complexity of IRC Solver). If the IRC solver stops at iteration $K$ with updated accuracy $\varepsilon_K$, then with probability at least $1 - \delta$ it converges, with the number of samples upper bounded by $n \leq \tilde{\mathcal{O}} \left( \frac{4 \gamma^2 R^2_{\max} SA}{(1-\gamma)^4 \varepsilon^2_K} \right)$, where $\tilde{\mathcal{O}}$ notation suppresses logarithmic terms.*

## 4.2 Sample Complexity of an ICRL Solver

Following a similar theoretical framework, Yue et al. (2024) derived the sample complexity for ICRL solver by defining the feasible cost set. We briefly recap and discuss the results below. Appendix B.6 provides a detailed review.

**Lemma 4.3.** *(Feasible cost set implicit (Yue et al., 2024, Lemma 4.3)). Under (Yue et al., 2024, Assumption 4.1), let $\mathfrak{P}$ be an ICI problem and $\mathbb{S}_{ICRL}$ be the ICRL solver, then $c$ is a feasible cost, i.e., $c \in \mathcal{C}_{\mathfrak{P}}$ if and only if $\forall (s,a) \in \mathcal{S} \times \mathcal{A}$:*

*(i) if $\pi^E(a|s) > 0$, $Q^{c,\pi^E}_{\mathcal{M} \cup c}(s,a) - V^{c,\pi^E}_{\mathcal{M} \cup c}(s) = 0$;*

*(ii) if $\pi^E(a|s) = 0$ and $A^{r,\pi^E}_{\mathcal{M} \cup c}(s,a) > 0$, $Q^{c,\pi^E}_{\mathcal{M} \cup c}(s,a) - V^{c,\pi^E}_{\mathcal{M} \cup c}(s) > 0$;*

*(iii) if $\pi^E(a|s) = 0$ and $A^{r,\pi^E}_{\mathcal{M} \cup c}(s,a) \leq 0$, $Q^{c,\pi^E}_{\mathcal{M} \cup c}(s,a) - V^{c,\pi^E}_{\mathcal{M} \cup c}(s) \leq 0$.*

*Remark* 4.4. Unlike the feasible reward correction terms in Lemma 4.1, the feasible cost function is closely related to the reward advantage function under an expert policy, i.e., $A^{r,\pi^E}_{\mathcal{M} \cup c}$. If $A^{r,\pi^E}_{\mathcal{M} \cup c}(s,a) > 0$, the action $a$ yields higher rewards than the expert. Such actions must violate the underlying constraints (case (ii)); otherwise, the expert policy could be further improved, contradicting its optimality.

Similarly, to compute the sample complexity of the ICRL solver, we utilize the uniform sampling strategy in Appendix Algorithm 1. Guaranteed by the strategy, the Hausdorff distance $d_H$ between

the ground truth and estimated feasible cost set, i.e., $\mathcal{C}_{\mathfrak{P}}$ and $\mathcal{C}_{\widehat{\mathfrak{P}}}$, is upper bounded by

$$d_H(\mathcal{C}_{\mathfrak{P}}, \mathcal{C}_{\widehat{\mathfrak{P}}}) \leq \max_{(s,a)\in\mathcal{S}\times\mathcal{A}} \mathcal{I}_{k+1}^c(s,a), \text{ with } \mathcal{I}_{k+1}^c(s,a) = \frac{\sigma}{(1-\gamma)^2}\sqrt{\frac{\ell_{k+1}(s,a)}{2N_{k+1}^+(s,a)}},$$

where $\sigma = \sqrt{3}\gamma C_{\max}\left(R_{\max}(3+\gamma)/\min^+\left|A_{\mathcal{M}\cup c}^{r,\pi^E}\right| + (1-\gamma)\right)$.

**Theorem 4.5.** *(Sample Complexity of ICRL solver (Yue et al., 2024, Theorem C.9)). If the ICRL solver terminates at iteration $K$ with the updated accuracy $\varepsilon_K$, then with probability at least $1 - \delta$, it converges with a number of samples upper bounded by $n \leq \tilde{\mathcal{O}}\left(\frac{\gamma^2\sigma^2 SA}{(1-\gamma)^6\varepsilon_K^2}\right)$.*

**Discussion.** By comparing Theorem 4.2 with Theorem 4.5, we observe that the sample complexity of ICRL solver exceeds that of IRC solver by a factor of $1/(1-\gamma)^2$. This increase stems from estimating the reward advantage function $A_{\mathcal{M}\cup c}^{r,\pi^E}$, which indictates whether to impose additional costs on any state-action pairs. Specifically, in the tri-level optimization in (1), establishing the feasible cost functions relies on the convergence of the Lagrange multiplier $\lambda$ to its optimum while $\lambda$ increases whenever $A_{\mathcal{M}\cup c}^{r,\pi^E}(s,a) > 0$ to penalize unsafe policies.

# 5 CROSS-ENVIRONMENT TRANSFERABILITY: GENERALIZING THE SAFETY AND OPTIMALITY OF CONSTRAINTS

Beyond cloning the behaviors of expert agents, a critical application of the inverse optimization methods (including IRL, ICRL, and IRC) is to infer generalizable oracle signals, such as rewards or constraints, that can effectively guide agent behaviors in similar environments. We use teal and purple colors to distinguish the source CMDP $\mathcal{M}_c = \mathcal{M} \cup c^E = (\mathcal{S}, \mathcal{A}, P_\mathcal{T}, r, c^E)$ and the target CMDP $\mathcal{M}_c' = \mathcal{M}' \cup (c')^E = (\mathcal{S}, \mathcal{A}, P_\mathcal{T}', r', (c')^E)$, where they can differ in the reward function, the transition model and underlying cost functions (i.e., $r' \neq r, P_\mathcal{T}' \neq P_\mathcal{T}$ and $c^E \neq (c')^E$). Under this setting, our study adheres to the standard ICI problem fashion, assuming access to a single expert within a specific environment. We investigate the transferability of learned constraints across source and target CMDPs, focusing on two key aspects: safety and optimality.

## 5.1 GENERALIZING SAFETY GUARANTEES ACROSS DIVERSE ENVIRONMENTS

Transferring a recovered reward correction term $\Delta r$ or a cost function $c$ introduces new challenges that remain largely unexplored in the ICI literature since there is no guarantee that $\Delta r$ or $c$ with the new reward function $r'$ and new transition model $P_\mathcal{T}'$ will make the new expert policy $(\pi')^E$ *constraint satisfying* in the target CMDP $\mathcal{M}_c'$. That is to say, transferred $\Delta r$ or $c$ may lead to unsafe policies (let alone optimal policies) in the target environment.

**Challenges in Ensuring Safety with IRC.** Although the IRC solver is more sample-efficient in training in the source environment, it struggles to guarantee safety in the target environment, as illustrated in Figure 1. Consider a hard constraint scenario; soft constraints will be discussed later. In the source environment, trajectory $\tau_1$ (with a reward $r_S(\tau_1) = 2$) has a larger reward than $\tau_2$ ($r_S(\tau_2) = 1$), but the expert agent prefers $\tau_2$ since $\tau_1$ passes through an unsafe state $s^c$. To align with the expert's demonstration, the IRC solver learns a feasible reward correction term $\Delta r(s^c) = -1 - \beta$ (where $\beta > 0$), ensuring that $\tau_1$ achieves lower rewards than

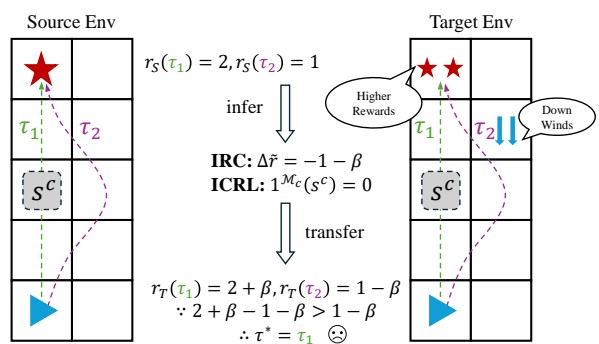

Figure 1: An example showing that transferring $\Delta r$ and constraint $\mathbb{1}^{\mathcal{M}_c}(s^c) = 0$ learned in the source environment (left) to the target environment (right) induces different optimal policies ($\tau_1$ denotes the optimal path and $\tau_2$ denotes the second-best path).

$\tau_2$: $r_S(\tau_1) + \Delta r(\tau_1) = 1 - \beta < 1 = r_S(\tau_2) + \Delta r(\tau_2)$ (note that $\Delta r(\tau_2) = 0$ since $\tau_2$ does not pass through $s^c$). However, when this correction term is transferred to a target environment—where reward functions and transition dynamics slightly differ ($r_T(\tau_1) = 2 + \beta$, $r_T(\tau_2) = 1 - \beta$)—it renders the unsafe path $\tau_1$ as deceptively safe since $\tau_1$ now achieves higher rewards than $\tau_2$: $r_T(\tau_1) + \Delta r(\tau_1) = 1 \geq 1 - \beta = r_T(\tau_2) + \Delta r(\tau_2)$. This misguides the agent to prefer $\tau_1$.

In essence, the reward correction terms reflect the extent of penalty applied to constraint-violating actions in the source environment. These penalties can be easily *offset* by increasing the gains on penalized actions or decreasing the gains on other actions in the target environment. Comparably, the ICRL solvers do not have such difficulty since they learn a hard constraint that invalidates any visit to $s^c$ (e.g., the feasibility $1^{\mathcal{M}_c}(s^c) = 0$). The cost function directly decides the boundaries of safe region, a modification that cannot be compensated for in the same manner as correction terms.

**A Formal Study of Transferability in Safety.** Transferability with single expert knowledge requires similarity restrictions between source and target environments (Metelli et al., 2021, Assumption 4.1) (Schlaginhaufen & Kamgarpour, 2024, Theorem 3.10). In the context of safety, we expect the learned constraint knowledge to remain at least partially active in the target environment, i.e., the two constrained regions overlap so that the learned constraint information can be effectively reused. This property is characterized as follows:

**Assumption 5.1.** (Similarity). Let $(\mathcal{M}, \pi^E)$ and $(\mathcal{M}', (\pi')^E)$ represent the source and target ICI problems. The intersection of their constraint-violating region is not empty: $\mathcal{G} = \{(s, a) \mid A^{r, \pi^E}_{\mathcal{M} \cup c^E}(s, a) > 0\} \cap \{(s, a) \mid A^{r', (\pi')^E}_{\mathcal{M}' \cup (c')^E}(s, a) > 0\} \neq \varnothing$.

Under this assumption, we begin by addressing the hard constraint scenario, in which the ICRL solver outputs a set of feasible cost functions, all of which can be safely transferred to the target CMDP, i.e., ensuring safety within $\mathcal{G}$. This property is illustrated in the following lemma:

**Lemma 5.2.** *Suppose a hard constraint scenario. For any $(s', a') \in \mathcal{G}$, the feasible cost function $\hat{c}$ inferred by the ICRL solver can prevent the visitation to $(s', a')$ in the target CMDP.*

However, the reward correction term learned by the IRC solver fails to guarantee safety in the target environment. In the following, we identify conditions under which $\Delta r$ is not transferable.

**Theorem 5.3.** *Suppose a hard constraint scenario. At state $s$, let $a^E$ denote the expert action, $a^C$ denote the action that satisfies $(s, a^C) \in \mathcal{G}$ and $a^O$ denote the other actions. $\forall r' \in [0, R_{\max}]^{\mathcal{S} \times \mathcal{A}}$ and $\forall P'_{\mathcal{T}} \in \Delta^{\mathcal{S}}_{\mathcal{S} \times \mathcal{A}}$, if $\exists s \in \mathcal{S}$, $\forall a^E, a^O \in \mathcal{A}$, $\exists a^C \in \mathcal{A}$ that satisfies the following conditions, then a reward correction term $\Delta r$ constructed by such Q-functions leads to unsafety in the target CMDP,*

$$Q^{r+\Delta r, \pi^E}_{\mathcal{M} \cup (r+\Delta r)}(\mathbf{e}_{(s,a^E)} - \mathbf{e}_{(s,a^C)}) \geq 0, Q^{r+\Delta r, \pi^E}_{\mathcal{M} \cup (r+\Delta r)}(\mathbf{e}_{(s,a^E)} - \mathbf{e}_{(s,a^O)}) \geq 0,$$

$$Q^{r+\Delta r, \pi^E}_{\mathcal{M} \cup (r+\Delta r)}(\mathbf{e}_{(s,a^E)} - \mathbf{e}_{(s,a^C)}) < (Y')^{-1} \left[ r' - r + (Y - Y')Q^{r+\Delta r, \pi^E}_{\mathcal{M} \cup (r+\Delta r)} \right] (\mathbf{e}_{(s,a^C)} - \mathbf{e}_{(s,a^E)}),$$

$$Q^{r+\Delta r, \pi^E}_{\mathcal{M} \cup (r+\Delta r)}(\mathbf{e}_{(s,a^O)} - \mathbf{e}_{(s,a^C)}) < (Y')^{-1} \left[ r' - r + (Y - Y')Q^{r+\Delta r, \pi^E}_{\mathcal{M} \cup (r+\Delta r)} \right] (\mathbf{e}_{(s,a^C)} - \mathbf{e}_{(s,a^O)}),$$

*where $Y = (I_{\mathcal{S} \times \mathcal{A}} - \gamma P_{\mathcal{T}} \pi^E)$, $Y' = (I_{\mathcal{S} \times \mathcal{A}} - \gamma P'_{\mathcal{T}} (\pi')^E)$ and $\mathbf{e}_{(s,a)}$ denotes a vector with value of 1 at index $(s, a)$ and 0 elsewhere.*

The first two inequalities, obtained from Lemma 4.1, ensure the optimality of the source expert policy $\pi^E$. The last two inequalities state that unsafe actions will be chosen if increments in the Q function of constraint-violating actions are larger than other actions after transfer. We provide a numerical validation of the above theorem in Appendix B.7, regarding the example in Figure 1.

**Extension to Soft Constraint.** The above results pertain to hard constraint scenarios. In cases of the soft constraint, i.e., threshold $\epsilon > 0$ in (PI), any $(s, a)$ can be visited by feasible policies because the cost of visiting $(s, a)$ can always be mitigated via the discount factor $\gamma < 1$. In this sense, cost functions, like reward correction terms, reflect the degree of penalization for constraint-violating actions that can be compensated by new transition dynamics and expert policies. Although the recovered cost function no longer guarantees safety in $\mathcal{G}$, it still outperforms inferred reward correction terms because it resists variations between source and target reward functions. We provide a detailed analysis for soft constraints in Appendix B.8.2.

## 5.2 GENERALIZING OPTIMALITY GUARANTEES ACROSS DIVERSE ENVIRONMENTS

The above analysis primarily emphasizes ensuring safety by satisfying constraints in the target environment. However, it does not guarantee that the resulting policy is optimal with respect to constrained reward maximization. In this context, a trivial solution involves blocking all state-action pairs deviating from expert trajectories. However, such an approach compromises the optimality of the resulting policy. Building upon safety guarantees, this section establishes the theoretical foundations for policy optimality in the target environment under the estimated *hard* constraint. As safety takes precedence over optimality, the IRC solver is excluded from this discussion.

**Transferability in Optimality for ICRL Solver.** Guaranteeing the optimality of transferred cost functions to target environments requires sufficient knowledge of the environment landscape. To achieve this goal, we recognize the difficulty of obtaining optimality guarantees under unregularized settings (Schlaginhaufen & Kamgarpour, 2024, Remark 3.12) and emphasize that exploration is crucial for obtaining theoretical guarantees of transferability.

**Assumption 5.4.** (Exploration). Let $\mathcal{F} = \{s \in \mathcal{S} \mid \exists a \in \mathcal{A}\colon c(s,a) = 0\}$ be the state feasible region, and $\mathcal{Q} = \{\mu \in \mathbb{R}_+^{\mathcal{S} \times \mathcal{A}}\colon (E - \gamma P)^\top \mu = (1-\gamma)\mu_0\} \subseteq \Delta^{\mathcal{S} \times \mathcal{A}}$ be the set of occupancy measures that satisfies the Bellman flow constraints. We assume that for any $s \in \mathcal{F}$ and $\mu \in \mathcal{Q}$, the state occupancy measure $\mu(s) := \sum_a \mu(s,a)$ is lower bounded by a positive constant, i.e., $\mu(s) \geq \mu_{\min} > 0$.

To ensure the policy can exhibit exploratory behavior, we adjust the CRL objective (PI) by assuming the optimal policy maximizes the regularized objective in the following:

$$\pi^* = \arg\max_\pi \; \mathbb{E}_{\mu_0, \pi, P_\mathcal{T}}\left[\sum_{t=0}^\infty \gamma^t \Big(r(s_t, a_t) - \lambda^* \cdot c(s_t, a_t) + h(\pi(\cdot|s_t))\Big)\right], \qquad (4)$$

where the Shannon entropy regularizer $h(\pi(\cdot|s)) := -\alpha\mathbb{E}_\pi[\log\pi(a|s)]$ and the weighting term $\alpha > 0$. Additionally, we model the influence of policy by deriving the occupancy measure $\mu$. As a result, the dual representation of the CRL problem (DI) can be recast as a convex optimization problem (Altman, 2021; Puterman, 2014):

$$\mathsf{CRL}_{P_\mathcal{T}}(r,c) := \arg\max_{\mu \in \mathcal{Q}} J(r, \lambda^*, c, \mu), \text{ with } J(r, \lambda^*, c, \mu) = \langle r - \lambda^* c, \mu\rangle - \mathbb{E}_\mu[h(\pi^\mu)], \quad (5)$$

where $\mu(s,a) = (1-\gamma)\mathbb{E}_{\pi, P_\mathcal{T}}\big[\sum_{t=0}^\infty \gamma^t \mathbb{1}(s_t = s, a_t = a)\big]$ and $\pi^\mu(a|s) = \mu(s,a)/\mu(s)$.

Under this formulation, to quantify the performance of a given occupancy measure $\mu$ under an inferred cost function $c$, we define the sub-optimality gap as follows:

$$\ell_{P_\mathcal{T}}^{r,\lambda^*}(c,\mu) := \max_{\mu' \in \mathcal{Q}} J(r, \lambda^*, c, \mu') - J(r, \lambda^*, c, \mu). \qquad (6)$$

Building upon this, we extend the transferability definition from Schlaginhaufen & Kamgarpour (2024, Definition 3.1) from regular MDP settings to CMDP settings.

**Definition 5.5.** ($\varepsilon$-transferability for cost). For some fixed $\varepsilon > 0$, we say the inferred cost function $\hat{c}$ is $\varepsilon$-transferable to some new reward function $r' \in [0, R_{\max}]^{\mathcal{S} \times \mathcal{A}}$ and transition law $P_\mathcal{T}' \in \Delta_\mathcal{S}^{\mathcal{S} \times \mathcal{A}}$ if $\ell_{P_\mathcal{T}'}^{r',(\lambda')^*}(\hat{c}, \mathsf{CRL}(c^E)) \leq \varepsilon$.

Intuitively, $\varepsilon$-transferability quantifies how well the inferred cost function $\hat{c}$ captures the preference of expert behaviors in a different environment. When the gap is large, it indicates that certain feasible state-action pairs are overly penalized, potentially leading to suboptimal policy decisions. Conversely, when the gap is small, the estimated cost function $\hat{c}$ accurately explains the expert's behaviors,

A key element in bounding this suboptimality gap in target environments is to constrain the similarity between source and target transition laws. To represent the feasible cost set $\mathcal{C}_{\mathbb{S}_{\mathrm{ICRL}}}$, we map a transition law $P_\mathcal{T}$ to a subspace $\mathcal{U}_{P_\mathcal{T}} := \mathrm{im}(E - \gamma P_\mathcal{T})$ via potential shaping transformation (Ng et al., 1999). A detailed discussion of this transformation for cost equivalence is provided in Appendix B.28. We utilize principal angles as a generalization of angles between higher-dimension planes to provide a more refined measure of similarity and dissimilarity between them.

**Definition 5.6.** (Principal angles (Galántai, 2013)) Let $\mathcal{V}, \mathcal{W} \subseteq \mathbb{R}^n$ be two subspaces of dimension $m \leq n$. The principal angles $0 \leq \theta_1(\mathcal{V}, \mathcal{W}) \leq \ldots \leq \theta_m(\mathcal{V}, \mathcal{W}) =: \theta_{\max(\mathcal{V},\mathcal{W})} \leq \pi/2$ between $\mathcal{V}$ and $\mathcal{W}$ are defined recursively via

$$\cos(\theta_i(\mathcal{V}, \mathcal{W})) = \max_{v \in \mathcal{V}, w \in \mathcal{W}} \langle v, w\rangle \text{ s.t. } \|v\|_2 = \|w\|_2 = 1, \; \langle v, v_j\rangle = \langle w, w_j\rangle = 0, \; j = 1, \ldots, i-1,$$

where $v_j, w_j$ are the maximizers corresponding to the angle $\theta_j$. For two transition laws $P_{\mathcal{T}1}, P_{\mathcal{T}2}$, we define $\theta_i(P_{\mathcal{T}1}, P_{\mathcal{T}2}) := \theta_i(\mathcal{U}_{P_{\mathcal{T}1}}, \mathcal{U}_{P_{\mathcal{T}2}})$ and $\theta_{\max}(P_{\mathcal{T}1}, P_{\mathcal{T}2}) := \theta_{\max}(\mathcal{U}_{P_{\mathcal{T}1}}, \mathcal{U}_{P_{\mathcal{T}2}})$.

**Theorem 5.7.** *Let $P'_{\mathcal{T}}$ be the target transition law, the source and target ground-truth constraints are the same $(c')^E = c^E$ and $d_1 = \|[c^E - \hat{c}]_{\mathcal{U}_{P'_{\mathcal{T}}}}\|_2$. Suppose that Assumption 5.4 holds. If $\ell_{P_{\mathcal{T}}}^{r',(\lambda')^*}(\hat{c}, \mathsf{CRL}(c^E)) \leq \varepsilon_1$, then $\hat{c}$ is $\varepsilon$-transferable to the target environment with*

$$\varepsilon = 2\max\left\{d_1^2 \sin\left(\theta_{\max}(P_{\mathcal{T}}', P_{\mathcal{T}})\right)^2 / 2, 2\varepsilon_1/\sigma_{\mathcal{R}}\right\}/\eta, \tag{7}$$

*where $\sigma_{\mathcal{R}}$ and $\eta$ are regularity constants, given in Appendix B.25.*

The theorem asserts that if the two transition laws are close and the recovered cost has a small suboptimality gap in the target environment, then $\varepsilon$-optimality of the recovered cost is guaranteed. More specifically, we observe that the transferability in optimality of cost is affected by two key factors. The first factor is the discrepancy between the source and target CMDP, which encompasses: 1) the difference in rewards and Lagrange multipliers, captured by $\varepsilon_1$, and 2) the discrepancy in transition dynamics, indicated by $\theta_{\max}$. The second factor is the estimation error associated with the recovered cost $\hat{c}$, denoted by $d_1$.

## 6 EMPIRICAL EVALUATION

We empirically evaluate the ICRL solver against the IRC solver in four different constrained Gridworld environments. For each Gridworld, source and target environments differ in reward functions and transition dynamics. Code is available

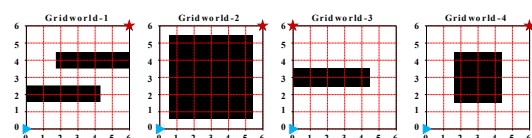

Figure 2: Four different Gridworld environments.

at `https://github.com/Bobyue0118/Constraint-Inference-in-Safe-IRL`.

**Experiment Setting.** We focus on evaluating the *training efficiency* and *transferability* of the ICRL and IRC solvers. The results are assessed using two key metrics:1) *discounted cumulative rewards*, which quantify the total rewards achieved by the learned policy. 2) *discounted cumulative costs*, which calculate the total costs incurred by the learned policy. We compare the uniform sampling strategy (Appendix Algorithm 1) of the ICRL and IRC solvers.

In the environment, the agent's goal is to navigate from a starting location (blue) to a target location (red) while avoiding constraints (black). The expert policy is trained under ground truth constraints. Four source environments exhibit a stochasticity of $p = 0.05$, i.e., the agent takes a uniformly randomized action with that probability, while four target environments feature a higher stochasticity of $p = 0.1$. All rewards are assigned at the target location, with identical values of 1 in four source environments and $2, 7, 7, 15$ in four respective target environments.

Figure 3 demonstrates four learned cost functions by the ICRL solver at each state (top) and four learned reward correction terms by the IRC solver at each state (bottom), in the source environments of four Gridworld settings (from left to right, Gridworld-1,2,3,4). Figure 4 demonstrates the accumulated rewards and costs of resulting policies learned under inferred reward correction terms by the IRC solver and cost functions by the ICRL solver at each iteration. In the source environments, we observe that the reward and cost curve of the IRC solver converges more quickly than the ICRL solver, indicating higher training efficiency. However, in the target environments, we find that inferred reward correction terms lead to unsafe policies (costs exceeding $0$), whereas recovered cost functions ensure both safety (costs converging to $0$) and optimality (rewards converging to the expert).

For continuous environments, we use the maximum entropy framework of ICRL (Malik et al., 2021) and simplified IRL framework for constraint inference (Hugessen et al., 2024) where two frameworks recover constraint knowledge that best explains the expert demonstrations from an offline dataset. We report additional experimental results and implementation details in Appendix C, D and E.

## 7 CONCLUSION

**Summary.** In this paper, we study the novel challenge of transferring learned constraint information from source to target environments. Constraint information can be either implicit reward correction

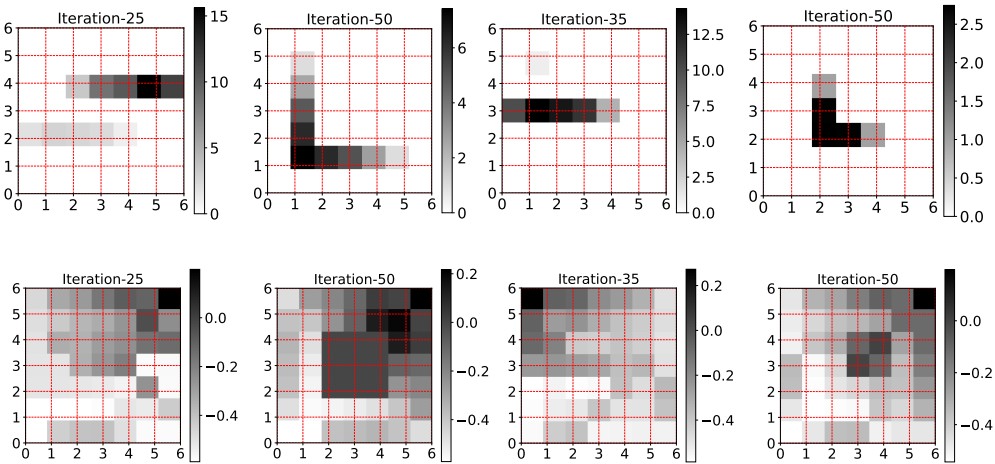

Figure 3: Learned constraint information by the ICRL solver (top) and the IRC solver (bottom).

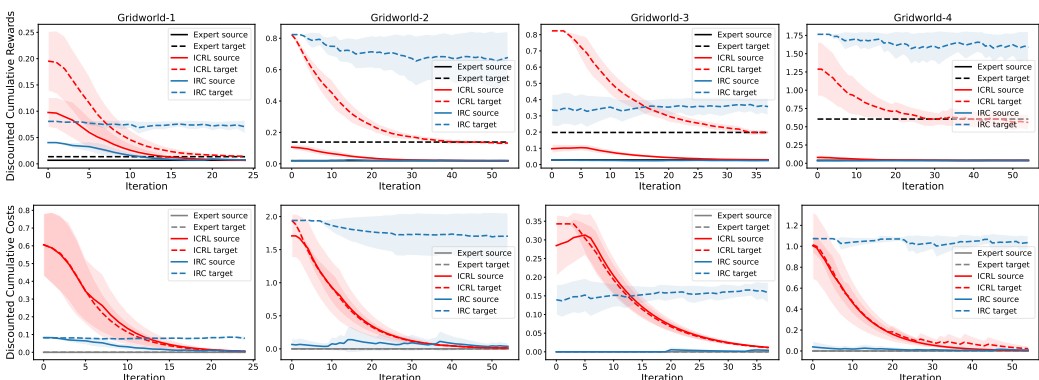

Figure 4: Training curves of discounted cumulative rewards (top), costs (bottom) for the ICRL (red) and IRC (blue) solvers across four Gridworld environments. The expert's rewards and costs are represented in grey. Solid and dashed lines correspond to source and target environments, respectively.

terms by IRC solvers or explicit cost functions by ICRL solvers. We compare both solvers regarding training efficiency and transferability. Although the IRC solver is guaranteed lower sample complexity, the additional sample complexity required by the ICRL solver proves crucial for ensuring both safety and optimality in target environments. Under hard constraints, the recovered cost functions strictly prevent the agent in target environments from entering overlapping constrained regions, whereas the reward correction terms can be easily offset by variations in transition laws and reward functions, leading to insufficient constraint representation. We also derive conditions that limit the similarity between source and target environments to ensure the optimality for the ICRL solver. Empirical studies across various environments validate our findings.

**Limitations and Future Work.** Our research initiates intriguing avenues for future studies. First, extending our approach to incorporate demonstrations from multiple experts, including sub-optimal ones, would be a valuable contribution to the field. Second, we believe that extensions to more complex and scalable tasks, particularly those in real-world applications, such as safety issues in robotics and Large Language Models, could offer valuable insights into the practical challenges and opportunities involved in transferring constraint information.

## ACKNOWLEDGMENTS

This work is supported in part by Shenzhen Fundamental Research Program (General Program) under grant JCYJ20230807114202005, Guangdong-Shenzhen Joint Research Fund under grant 2023A1515110617, Guangdong Basic and Applied Basic Research Foundation under grant 2024A1515012103, and Guangdong Provincial Key Laboratory of Mathematical Foundations for Artificial Intelligence (2023B1212010001). Poupart and Gaurav were also supported by the Canada CIFAR AI Chair program, a discovery grant from the Natural Sciences and Engineering Research Council of Canada and a grant from IITP & MSIT of Korea (No. RS-2024-00457882, AI Research Hub Project).

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

## Table of Contents

## A  GENERAL NOTATIONS AND NOMENCLATURE

In Table 1, we report the general notations and nomenclature used in our paper.

## B  DEFINITIONS, THEOREMS, AND PROOFS

### B.1  ADDITIONAL DEFINITIONS

**Definition B.1.** (Hausdorff distance). (Rockafellar & Wets, 2009). Let $(M, d)$ be a metric space. The Hausdorff distance $d_H$ between two non-empty subsets $A \subseteq M$ and $B \subseteq M$ with distance function $d$ is defined as:

$$d_H(A, B) = \max \left\{ \sup_{a \in A} \inf_{b \in B} d(a, b), \sup_{b \in B} \inf_{a \in A} d(b, a) \right\}$$

where:

- $\inf_{b \in B} d(a, b)$ denotes the shortest distance from a point $a \in A$ to all points in set $B$.

- $\sup_{a \in A} \inf_{b \in B} d(a, b)$ is the maximum of these shortest distances for all points in $A$.

- Similarly, $\sup_{b \in B} \inf_{a \in A} d(b, a)$ represents the maximum shortest distance for all points in $B$ to set $A$.

**Definition B.2.** (Operators). Let $f \in \mathbb{R}^{\mathcal{S}}$ and $g \in \mathbb{R}^{\mathcal{S} \times \mathcal{A}}$. We abuse $P_{\mathcal{T}}$ and $\pi$ as the operators induced by the transition model $p$ and by the policy $\pi$, i.e.,$(P_{\mathcal{T}} f)(s, a) = \sum_{s' \in \mathcal{S}} P_{\mathcal{T}}(s'|s, a) f(s')$ and $(\pi g)(s) = \sum_{a \in \mathcal{A}} \pi(a|s) g(s, a)$. Moreover, the expansion operator $(Ef)(s, a) = f(s)$. Given $\pi \in \Delta_{\mathcal{S}}^{\mathcal{A}}$, we denote with $(B^{\pi} g)(s, a) = g(s, a) \mathbb{1}\{\pi(a|s) > 0\}$ and $(\overline{B}^{\pi} g)(s, a) = g(s, a) \mathbb{1}\{\pi(a|s) = 0\}$.

Table 1: General Notations and Nomenclature

| Symbol | Name | Signature |
|--------|------|-----------|
| $\mathcal{S}$ | State space | / |
| $\mathcal{A}$ | Action space | / |
| $r/r'$ | Source/Target reward function | $\mathbb{R}^{\mathcal{S}\times\mathcal{A}}$ |
| $P_{\mathcal{T}}/P'_{\mathcal{T}}$ | Source/Target transition model | $\Delta^{\mathcal{S}}_{\mathcal{S}\times\mathcal{A}}$ |
| $\pi^E/(\pi')^E$ | Source/Target expert policy | $\Delta^{\mathcal{A}}_{\mathcal{S}}$ |
| $c^E$ | Expert cost function | $\mathbb{R}^{\mathcal{S}\times\mathcal{A}}$ |
| $c$ | Cost function | $\mathbb{R}^{\mathcal{S}\times\mathcal{A}}$ |
| $\Delta r$ | Reward correction term | $\mathbb{R}^{\mathcal{S}\times\mathcal{A}}$ |
| $Q^{r,\pi}_{\mathcal{M}}$ | Reward action-value function for $r$ of $\pi$ in $\mathcal{M}$ | $\mathbb{R}^{\mathcal{S}\times\mathcal{A}}$ |
| $V^{r,\pi}_{\mathcal{M}}$ | Reward state-value function for $r$ of $\pi$ in $\mathcal{M}$ | $\mathbb{R}^{\mathcal{S}}$ |
| $A^{r,\pi}_{\mathcal{M}}$ | Reward advantage function for $r$ of $\pi$ in $\mathcal{M}$ | $\mathbb{R}^{\mathcal{S}\times\mathcal{A}}$ |
| $Q^{c,\pi}_{\mathcal{M}_c}$ | Cost action-value function for $c$ of $\pi$ in $\mathcal{M}_c$ | $\mathbb{R}^{\mathcal{S}\times\mathcal{A}}$ |
| $V^{c,\pi}_{\mathcal{M}_c}$ | Cost state-value function for $c$ of $\pi$ in $\mathcal{M}_c$ | $\mathbb{R}^{\mathcal{S}}$ |
| $Q^{r,\pi}_{\mathcal{M}_c}$ | Reward action-value function for $r$ of $\pi$ in $\mathcal{M}_c$ | $\mathbb{R}^{\mathcal{S}\times\mathcal{A}}$ |
| $V^{r,\pi}_{\mathcal{M}_c}$ | Reward state-value function for $r$ of $\pi$ in $\mathcal{M}_c$ | $\mathbb{R}^{\mathcal{S}}$ |
| $A^{r,\pi}_{\mathcal{M}_c}$ | Reward advantage function for $r$ of $\pi$ in $\mathcal{M}_c$ | $\mathbb{R}^{\mathcal{S}\times\mathcal{A}}$ |
| $\mathfrak{P}$ | ICI problem $(\mathcal{M}, \pi^E)$ | / |
| $\mathbb{S}_{\text{IRC}}$ | IRC solver | / |
| $\mathcal{R}_{\mathfrak{P}}$ | Set of feasible reward correction terms | $\{\mathbb{R}^{\mathcal{S}\times\mathcal{A}}\}$ |
| $\mathbb{S}_{\text{ICRL}}$ | ICRL solver | / |
| $\mathcal{C}_{\mathfrak{P}}$ | Set of feasible cost functions | $\{\mathbb{R}^{\mathcal{S}\times\mathcal{A}}\}$ |
| $\varepsilon$ | Target accuracy | $\mathbb{R}^+$ |
| $\delta$ | Significancy | $(0,1)$ |
| $\theta$ | Principal angle | $[0, \pi/2]$ |
| $\|\cdot\|_\infty$ | Infinity norm | / |
| $E$ | Expansion operator | $\mathbb{R}^{\mathcal{S}} \to \mathbb{R}^{\mathcal{S}\times\mathcal{A}}$ |
| $\text{im}(T)$ | Image of a linear operator $T$ | / |

**Definition B.3.** (Image of a linear operator). Let $T : V \to W$ be a linear operator from vector space $V$ to vector space $W$. The image of $T$, denoted as $\text{im}(T)$, is defined as:

$$\text{im}(T) = \{T(v) \mid v \in V\}.$$

This is the set of all vectors in $W$ that can be written as $T(v)$ for some $v \in V$. The image of $T$ is a subspace of $W$.

**Definition B.4.** (Inifinity norm). For a vector $a$, we define the vector infinity norm as $\|a\|_\infty = \max_i |a_i|$. For a matrix $M$, we define the matrix infinity norm as $\|M\|_\infty = \max_i \sum_j |M_{ij}|$.

## B.2 Estimating the Transition Model and Expert Policy

We define how we estimate the transition model and the expert policy with a generative model in Algorithm 1. When the generative model is queried about a state-action pair $(s, a)$, it responds with a transition triple $(s, a, s')$ and an expert action $a^E \sim \pi^E(s)$

In each iteration of uniform sampling, we query the generative model with each state-action pair multiple times. At iteration $k$, we denote by $n_k(s, a, s')$ the number of times we observe the transition tuple $(s, a, s')$. We further denote $n_k(s, a) = \sum_{s' \in \mathcal{S}} n_k(s, a, s')$. At iteration $k$, we denote by $n_k^E(s, a)$ the number of times we observe action $a$ as an expert decision at state $s$. We further denote $n_k^E(s) = \sum_{a \in \mathcal{A}} n_k^E(s, a)$.

We define four cumulative counts from iteration 1 to $k$ as $N_k(s, a, s') = \sum_{j=1}^{k} n_j(s, a, s')$ and $N_k(s, a) = \sum_{j=1}^{k} n_j(s, a)$, $N_k^E(s, a) = \sum_{j=1}^{k} n_j^E(s, a)$ and $N_k^E(s) = \sum_{j=1}^{k} n_j^E(s)$. Finally, the

estimated transition model and the estimated expert policy are defined as:

$$\widehat{P_{\mathcal{T}}}_k(s'|s,a) = \frac{N_k(s,a,s')}{N_k^+(s,a)}, \quad \widehat{\pi}_k^E(a|s) = \frac{N_k^E(s,a)}{N_k^{E^+}(s)}, \tag{8}$$

where $x^+ = \max\{1, x\}$.

## B.3 Uniform Sampling Strategy for Each Solver

To acquire desired information, the agent utilizes the uniform sampling strategy to query a generative model. Specifically, the agent can always query a generative model about a state-action pair $(s, a)$ to receive a next state $s' \sim P(\cdot|s, a)$ and an expert action $a^E \sim \pi^E(\cdot|s)$.

---

**Algorithm 1** Uniform Sampling Strategy With a Generative Model

---

**Input:** significance $\delta$, target accuracy $\varepsilon$, maximum number of samples per iteration $n_{\max}$
Initialize $k \leftarrow 0$, $\varepsilon_0 = \frac{1}{1-\gamma}$
**while** $\varepsilon_k > \varepsilon$ **do**
    Collect $\lceil \frac{n_{\max}}{SA} \rceil$ samples from each $(s, a) \in \mathcal{S} \times \mathcal{A}$
    For IRC solver, update accuracy $\varepsilon_{k+1} = \frac{1}{1-\gamma} \max\limits_{(s,a) \in \mathcal{S} \times \mathcal{A}} \mathcal{I}_{k+1}^{\Delta r}(s, a)$
    For ICRL solver, update accuracy $\varepsilon_{k+1} = \frac{1}{1-\gamma} \max\limits_{(s,a) \in \mathcal{S} \times \mathcal{A}} \mathcal{I}_{k+1}^{c}(s, a)$
    Update $\widehat{P_{\mathcal{T}}}_{k+1}$ and $\widehat{\pi}_{k+1}^E$ according to Eq. (8)
    $k \leftarrow k + 1$
**end while**

---

## B.4 Theoretical Results of IRL Solver

In this part, we reformulate the IRL solver for ICI based on Metelli et al. (2021).

**Inverse Reinforcement Learning (IRL).** Typically, IRL solvers recover the reward function from expert demonstrations, where the environment is always considered to be safe. To employ IRL for constraint inference in CMDP, we modify the original formalization of the IRL problem (Ng et al., 2000) as follows:

**Definition B.5.** (*IRL solver for ICI*). An IRL solver is denoted as $\mathbb{S}_{\text{IRL}}$. A corrected reward function $r^c$ is a *feasible* solution for an ICI problem $\mathfrak{P}$ if and only if $\pi^E$ is an optimal policy for $\mathcal{M} \cup r^c$. We denote by $\mathfrak{R}_{\mathfrak{P}}$ the set of feasible corrected reward functions.

**Lemma B.6.** (*Feasible reward set implicit*) (*Metelli et al., 2021, Lemma 3.1*). *Let $\mathfrak{P}$ be an ICI problem and $\mathbb{S}_{\text{IRL}}$ be the IRL solver. $r^c$ is a feasible corrected reward function, i.e, $r^c \in \mathfrak{R}_{\mathfrak{P}}$ if and only if $\forall (s, a) \in \mathcal{S} \times \mathcal{A}$:*

- *if $\pi^E(a|s) > 0$, $Q_{\mathcal{M} \cup r^c}^{r^c, \pi^E}(s, a) = V_{\mathcal{M} \cup r^c}^{r^c, \pi^E}(s)$,*

- *if $\pi^E(a|s) = 0$, $Q_{\mathcal{M} \cup r^c}^{r^c, \pi^E}(s, a) \leq V_{\mathcal{M} \cup r^c}^{r^c, \pi^E}(s)$.*

**Lemma B.7.** (*Metelli et al., 2021, Lemma B.1*). *Let $\mathfrak{P}$ be an ICI problem and $\mathbb{S}_{\text{IRL}}$ be the IRL solver. A Q-function satisfies the condition of Lemma B.6 if and only if there exist $\zeta \in \mathbb{R}_{\geq 0}^{\mathcal{S} \times \mathcal{A}}$ and $V^r \in \mathbb{R}^{\mathcal{S}}$ such that:*

$$Q_{\mathcal{M} \cup r^c}^{r^c, \pi^E} = -\overline{B}^{\pi^E} \zeta + EV^r. \tag{9}$$

*Furthermore, $\|V^r\|_\infty \leq \|Q_{\mathcal{M} \cup r^c}^r\|_\infty$ and the the expansion operator $E$ satisfies $(Ef)(s, a) = f(s)$.*

**Lemma B.8.** (*Feasible reward set explicit* (*Metelli et al., 2021, Lemma 3.2*)). *Let $\mathfrak{P}$ be an ICI problem and $\mathbb{S}_{\text{IRL}}$ be the IRL solver. A reward function $r^c \in \mathfrak{R}_{\mathfrak{P}}$ if and only if there exist $\zeta \in \mathbb{R}_{\geq 0}^{\mathcal{S} \times \mathcal{A}}$ and $V^r \in \mathcal{R}^{\mathcal{S}}$ such that*

$$r^c = -\overline{B}^{\pi^E} \zeta + (E - \gamma P_{\mathcal{T}}) V^r. \tag{10}$$

**Lemma B.9.** *(Error Propagation ([Metelli et al., 2021](), Theorem 3.1)). Let* $\mathfrak{P} = (\mathcal{M}, \pi^E)$ *and* $\widehat{\mathfrak{P}} = (\widehat{\mathcal{M}}, \widehat{\pi}^E)$ *be two ICI problems. Then, for any* $r^c \in \mathfrak{R}_{\mathfrak{P}}$ *such that* $r^c = -\overline{B}^{\pi^E}\zeta + (E - \gamma P_{\mathcal{T}})V^r$ *and* $\|r^c\|_\infty \leq R_{\max}$*, there exists* $\widehat{r}^c \in \mathfrak{R}_{\widehat{\mathfrak{P}}}$ *such that element-wise it holds that:*

$$|r^c - \widehat{r}^c| \leq \overline{B}^{\pi^E} B^{\widehat{\pi}^E}\zeta + \gamma \left| \left( P_{\mathcal{T}} - \widehat{P_{\mathcal{T}}} \right) V^r \right|. \tag{11}$$

*Furthermore,* $\|\zeta\|_\infty \leq \frac{R_{\max}}{1-\gamma}$ *and* $\|V^r\|_\infty \leq \frac{R_{\max}}{1-\gamma}$.

**Theorem B.10.** *(Sample Complexity of the IRL Solver ([Metelli et al., 2021](), Theorem 5.1)). If the IRL solver stops at iteration* $K$ *with accuracy* $\varepsilon_K$*, then with probability at least* $1 - \delta$ *it converges, with a number of samples upper bounded by:*

$$n \leq \tilde{\mathcal{O}} \left( \frac{\gamma^2 R_{\max}^2 SA}{(1-\gamma)^4 \varepsilon_K^2} \right). \tag{12}$$

*Proof.* The above result can be obtained by setting $\mathcal{M}' = \mathcal{M}$ and $\gamma' = \gamma$ in ([Metelli et al., 2021](), Theorem 5.1). □

*Remark* B.11. In IRL solvers, although $r$ in the CMDP is independent of the cost function $c$, $r^c$ indeed captures the information of underlying constraint signals for enabling the imitating agent to reproduce $\pi^E$. However, the IRL solver lacks a mechanism to leverage the known reward function $r$ for constraint inference. Moreover, since $r^c$ relies on the current reward function, directly transferring $r^c$ to another CMDP with a different reward function does not make sense. Therefore, it is necessary to adjust the setting of IRL solver to accommodate potential changes in the reward function. [Li et al.]() ([2023]()) propose modifying imperfect reward functions to align them with expert behaviors, but this approach lacks tractable sample complexity. To address these limitations, we propose a modified version of the IRL solver from a feasible set perspective, namely the Inverse Reward Correction (IRC) solver in the main paper.

## B.5 THEORETICAL RESULTS OF IRC SOLVER

In this part, we provide additional results regarding the IRC solver for ICI.

**Inverse Reward Correction (IRC).** Based on the known $r$, IRC solvers learn a reward correction term $\Delta r$ to capture constraint signals. The goal of IRC solvers is to enable the imitating agent to match expert demonstrations by following the corrected rewards: $r^c(s,a) = r(s,a) + \Delta r(s,a), \forall (s,a) \in \mathcal{S} \times \mathcal{A}$.

**Lemma 4.1.** *(Feasible reward correction set implicit). Let* $\mathfrak{P}$ *be an ICI problem and* $\mathbb{S}_{IRC}$ *be the IRC solver.* $\Delta r$ *is a feasible reward correction term, i.e.,* $\Delta r \in \mathcal{R}_{\mathfrak{P}}$ *if and only if* $\forall (s,a) \in \mathcal{S} \times \mathcal{A}$*:*

*(i) if* $\pi^E(a|s) > 0$, $Q_{\mathcal{M}\cup(r+\Delta r)}^{r+\Delta r, \pi^E}(s,a) = V_{\mathcal{M}\cup(r+\Delta r)}^{r+\Delta r, \pi^E}(s)$,

*(ii) if* $\pi^E(a|s) = 0$, $Q_{\mathcal{M}\cup(r+\Delta r)}^{r+\Delta r, \pi^E}(s,a) \leq V_{\mathcal{M}\cup(r+\Delta r)}^{r+\Delta r, \pi^E}(s)$.

*Proof.* From Lemma [B.6]() and the decomposition that $r^c(s,a) = r(s,a) + \Delta r(s,a), \forall (s,a)$, we have,

(i) if $\pi^E(a|s) > 0$, $Q_{\mathcal{M}\cup(r+\Delta r)}^{r+\Delta r, \pi^E}(s,a) = V_{\mathcal{M}\cup(r+\Delta r)}^{r+\Delta r, \pi^E}(s)$,

(ii) if $\pi^E(a|s) = 0$, $Q_{\mathcal{M}\cup(r+\Delta r)}^{r+\Delta r, \pi^E}(s,a) \leq V_{\mathcal{M}\cup(r+\Delta r)}^{r+\Delta r, \pi^E}(s)$.

□

**Corollary B.12.** *(Feasible reward correction set implicit). Let* $\mathfrak{P}$ *be an ICI problem and* $\mathbb{S}_{IRC}$ *be the IRC solver.* $\Delta r$ *is a feasible reward correction term, i.e.,* $\Delta r \in \mathcal{R}_{\mathfrak{P}}$ *if and only if for all* $(s,a) \in \mathcal{S} \times \mathcal{A}$*, it holds that:*

*(i) if* $\pi^E(a|s) > 0$, $Q_{\mathcal{M}\cup(r+\Delta r)}^{\Delta r, \pi^E}(s,a) = -A_{\mathcal{M}\cup(r+\Delta r)}^{r, \pi^E}(s,a) + V_{\mathcal{M}\cup(r+\Delta r)}^{\Delta r, \pi^E}(s)$,

*(ii) if* $\pi^E(a|s) = 0$, $Q_{\mathcal{M}\cup(r+\Delta r)}^{\Delta r, \pi^E}(s,a) \leq -A_{\mathcal{M}\cup(r+\Delta r)}^{r, \pi^E}(s,a) + V_{\mathcal{M}\cup(r+\Delta r)}^{\Delta r, \pi^E}(s)$.

*Proof.* Note that, for any given policy $\pi$, Q-function and V-function are both linear towards reward function, i.e., $Q^{r_1+r_2,\pi}_{\mathcal{M}\cup(r_1+r_2)} = Q^{r_1,\pi}_{\mathcal{M}\cup(r_1+r_2)} + Q^{r_2,\pi}_{\mathcal{M}\cup(r_1+r_2)}$ and $V^{r_1+r_2,\pi}_{\mathcal{M}\cup(r_1+r_2)} = V^{r_1,\pi}_{\mathcal{M}\cup(r_1+r_2)} + V^{r_2,\pi}_{\mathcal{M}\cup(r_1+r_2)}$. Thus, inheriting from Lemma 4.1, we obtain,

(i) if $\pi^E(a|s) > 0$, $\left[Q^{\Delta r,\pi^E}_{\mathcal{M}\cup(r+\Delta r)} + Q^{r,\pi^E}_{\mathcal{M}\cup(r+\Delta r)}\right](s,a) = \left[V^{r,\pi^E}_{\mathcal{M}\cup(r+\Delta r)} + V^{\Delta r,\pi^E}_{\mathcal{M}\cup(r+\Delta r)}\right](s)$,

(ii) if $\pi^E(a|s) = 0$, $\left[Q^{\Delta r,\pi^E}_{\mathcal{M}\cup(r+\Delta r)} + Q^{r,\pi^E}_{\mathcal{M}\cup(r+\Delta r)}\right](s,a) \leq \left[V^{r,\pi^E}_{\mathcal{M}\cup(r+\Delta r)} + V^{\Delta r,\pi^E}_{\mathcal{M}\cup(r+\Delta r)}\right](s)$.

By simple transposition, we derive,

(i) if $\pi^E(a|s) > 0$,
$$Q^{\Delta r,\pi^E}_{\mathcal{M}\cup(r+\Delta r)}(s,a) = -Q^{r,\pi^E}_{\mathcal{M}\cup(r+\Delta r)}(s,a) + V^{r,\pi^E}_{\mathcal{M}\cup(r+\Delta r)}(s) + V^{\Delta r,\pi^E}_{\mathcal{M}\cup(r+\Delta r)}(s),$$

(ii) if $\pi^E(a|s) = 0$,
$$Q^{\Delta r,\pi^E}_{\mathcal{M}\cup(r+\Delta r)}(s,a) \leq -Q^{r,\pi^E}_{\mathcal{M}\cup(r+\Delta r)}(s,a) + V^{r,\pi^E}_{\mathcal{M}\cup(r+\Delta r)}(s) + V^{\Delta r,\pi^E}_{\mathcal{M}\cup(r+\Delta r)}(s).$$

By the definition of advantage function $A^{r,\pi}_{\mathcal{M}\cup r_1}(s,a) = Q^{r,\pi}_{\mathcal{M}\cup r_1}(s,a) - V^{r,\pi}_{\mathcal{M}\cup r_1}(s)$, we obtain,

(i) if $\pi^E(a|s) > 0$, $Q^{\Delta r,\pi^E}_{\mathcal{M}\cup(r+\Delta r)}(s,a) = -A^{r,\pi^E}_{\mathcal{M}\cup(r+\Delta r)}(s,a) + V^{\Delta r,\pi^E}_{\mathcal{M}\cup(r+\Delta r)}(s)$,

(ii) if $\pi^E(a|s) = 0$, $Q^{\Delta r,\pi^E}_{\mathcal{M}\cup(r+\Delta r)}(s,a) \leq -A^{r,\pi^E}_{\mathcal{M}\cup(r+\Delta r)}(s,a) + V^{\Delta r,\pi^E}_{\mathcal{M}\cup(r+\Delta r)}(s)$.

$\square$

**Lemma B.13.** *Let $\mathfrak{P}$ be an ICI problem and $\mathbb{S}_{IRC}$ be the IRC solver. A Q-function satisfies the condition of Lemma 4.1 if and only if there exist $\zeta \in \mathbb{R}^{\mathcal{S}\times\mathcal{A}}_{\geqslant 0}$ and $V^r \in \mathbb{R}^{\mathcal{S}}$ such that:*

$$Q^{\Delta r,\pi^E}_{\mathcal{M}\cup(r+\Delta r)} = -\overline{B}^{\pi^E}\zeta - A^{r,\pi^E}_{\mathcal{M}\cup(r+\Delta r)} + EV^r. \tag{13}$$

*Proof.* The proof can be easily derived from from (Metelli et al., 2021, Lemma B.1). $\square$

**Lemma B.14.** *(Feasible reward correction set explicit). Let $\mathfrak{P}$ be an ICI problem and $\mathbb{S}_{IRC}$ be the IRC solver. $\Delta r$ is a feasible reward correction term, i.e., $\Delta r \in \mathcal{R}_{\mathfrak{P}}$ if and only if there exist $\zeta \in \mathbb{R}^{\mathcal{S}\times\mathcal{A}}_{\geq 0}$ and $V^r \in \mathbb{R}^{\mathcal{S}}$ such that:*

$$\Delta r = -\overline{B}^{\pi^E}\zeta - r + (E - \gamma P_{\mathcal{T}})V^{r,\pi^E}_{\mathcal{M}\cup(r+\Delta r)} + (E - \gamma P_{\mathcal{T}})V^r. \tag{14}$$

*Proof.* From Bellman equation (Sutton & Barto, 2018), we have,

$$\Delta r = \left(I_{\mathcal{S}\times\mathcal{A}} - \gamma P_{\mathcal{T}}\pi^E\right)Q^{\Delta r,\pi^E}_{\mathcal{M}\cup(r+\Delta r)},$$

$$r = \left(I_{\mathcal{S}\times\mathcal{A}} - \gamma P_{\mathcal{T}}\pi^E\right)Q^{r,\pi^E}_{\mathcal{M}\cup(r+\Delta r)}.$$

Thus, we obtain,

$$\Delta r = \left(I_{\mathcal{S}\times\mathcal{A}} - \gamma P_{\mathcal{T}}\pi^E\right)Q^{\Delta r,\pi^E}_{\mathcal{M}\cup(r+\Delta r)}$$

$$= \left(I_{\mathcal{S}\times\mathcal{A}} - \gamma P_{\mathcal{T}}\pi^E\right)\left(-\overline{B}^{\pi^E}\zeta - A^{r,\pi^E}_{\mathcal{M}\cup(r+\Delta r)} + EV^r\right)$$

$$= -\overline{B}^{\pi^E}\zeta - A^{r,\pi^E}_{\mathcal{M}\cup(r+\Delta r)} + EV^r + \gamma P_{\mathcal{T}}\pi^E A^{r,\pi^E}_{\mathcal{M}\cup(r+\Delta r)} - \gamma P_{\mathcal{T}}V^r$$

$$= -\overline{B}^{\pi^E}\zeta - \left(I_{\mathcal{S}\times\mathcal{A}} - \gamma P_{\mathcal{T}}\pi^E\right)A^{r,\pi^E}_{\mathcal{M}\cup(r+\Delta r)} + (E - \gamma P_{\mathcal{T}})V^r$$

$$= -\overline{B}^{\pi^E}\zeta - \left(I_{\mathcal{S}\times\mathcal{A}} - \gamma P_{\mathcal{T}}\pi^E\right)\left(Q^{r,\pi^E}_{\mathcal{M}\cup(r+\Delta r)} - EV^{r,\pi^E}_{\mathcal{M}\cup(r+\Delta r)}\right) + (E - \gamma P_{\mathcal{T}})V^r$$

$$= -\overline{B}^{\pi^E}\zeta - \left(I_{\mathcal{S}\times\mathcal{A}} - \gamma P_{\mathcal{T}}\pi^E\right)Q^{r,\pi^E}_{\mathcal{M}\cup(r+\Delta r)} + \left(I_{\mathcal{S}\times\mathcal{A}} - \gamma P_{\mathcal{T}}\pi^E\right)EV^{r,\pi^E}_{\mathcal{M}\cup(r+\Delta r)} + (E - \gamma P_{\mathcal{T}})V^r$$

$$= -\overline{B}^{\pi^E}\zeta - r + (E - \gamma P_{\mathcal{T}})V^{r,\pi^E}_{\mathcal{M}\cup(r+\Delta r)} + (E - \gamma P_{\mathcal{T}})V^r, \tag{15}$$

where the last equality utilizes $\pi^E E = I_{\mathcal{S}}$. $\square$

**Lemma B.15.** *(Error Propagation). Let $\mathfrak{P} = (\mathcal{M}, \pi^E)$ and $\widehat{\mathfrak{P}} = (\widehat{\mathcal{M}}, \widehat{\pi}^E)$ be two ICI problems. Then, for any $\Delta r \in \mathcal{R}_{\mathfrak{P}}$ such that $\Delta r = -\overline{B}^{\pi^E}\zeta - r + (E - \gamma P_{\mathcal{T}}) V^{r,\pi^E}_{\mathcal{M}\cup(r+\Delta r)} + (E - \gamma P_{\mathcal{T}}) V^r$, there exists $\widehat{\Delta r} \in \mathcal{R}_{\widehat{\mathfrak{P}}}$ such that element-wise it holds that:*

$$\left| \Delta r - \widehat{\Delta r} \right| \leq \overline{B}^{\pi^E} B^{\widehat{\pi}^E}\zeta + \gamma \left| \left( P_{\mathcal{T}} - \widehat{P_{\mathcal{T}}} \right) \left( V^{r,\pi^E}_{\mathcal{M}\cup(r+\Delta r)} + V^r \right) \right|. \tag{16}$$

*Furthermore, $\|\zeta\|_\infty \leq \frac{2R_{\max}}{1-\gamma}, \|V^{r,\pi^E}_{\mathcal{M}\cup(r+\Delta r)}\|_\infty \leq \frac{R_{\max}}{1-\gamma}$ and $\|V^r\|_\infty \leq \frac{R_{\max}}{1-\gamma}$.*

*Proof.* From Lemma B.14, we can express $\Delta r$ and $\widehat{\Delta r}$ as,

$$\Delta r = -\overline{B}^{\pi^E}\zeta - r + (E - \gamma P_{\mathcal{T}}) V^{r,\pi^E}_{\mathcal{M}\cup(r+\Delta r)} + (E - \gamma P_{\mathcal{T}}) V^r \tag{17}$$

$$\widehat{\Delta r} = -\overline{B}^{\widehat{\pi}^E}\widehat{\zeta} - r + (E - \gamma\widehat{P_{\mathcal{T}}}) V^{r,\widehat{\pi}^E}_{\widehat{\mathcal{M}}\cup(r+\Delta r)} + (E - \gamma\widehat{P_{\mathcal{T}}})\widehat{V}^r \tag{18}$$

Since we look for the existence of $\widehat{\Delta r}$, we provide a specific choice of $\widehat{\zeta}$ and $\widehat{V}^r$: $\widehat{\zeta} = \overline{B}^{\pi^E}\zeta$ and $\widehat{V}^r = V^r + V^{r,\pi^E}_{\mathcal{M}\cup(r+\Delta r)} - V^{r,\widehat{\pi}^E}_{\widehat{\mathcal{M}}\cup(r+\Delta r)}$

$$\Delta r - \widehat{\Delta r} = -\overline{B}^{\pi^E} B^{\widehat{\pi}^E}\zeta - \gamma \left( P_{\mathcal{T}} - \widehat{P_{\mathcal{T}}} \right) \left( V^{r,\pi^E}_{\mathcal{M}\cup(r+\Delta r)} + V^r \right) \tag{19}$$

By taking the absolute value and applying the triangular inequality, we obtain:

$$\left| \Delta r - \widehat{\Delta r} \right| \leq \overline{B}^{\pi^E} B^{\widehat{\pi}^E}\zeta + \gamma \left| \left( P_{\mathcal{T}} - \widehat{P_{\mathcal{T}}} \right) \left( V^{r,\pi^E}_{\mathcal{M}\cup(r+\Delta r)} + V^r \right) \right|. \tag{20}$$

Note that the $L_\infty$-norms of value function $\|V^{r,\pi^E}_{\mathcal{M}\cup(r+\Delta r)}\|_\infty \leq \frac{R_{\max}}{1-\gamma}$ and $\|V^r\|_\infty \leq \frac{R_{\max}}{1-\gamma}$. Then, by Lemma B.13, $\|\zeta\|_\infty \leq \frac{2R_{\max}}{1-\gamma}$. $\qquad\square$

**Theorem 4.2.** *(Sample Complexity of the IRC Solver). If an IRC solver stops at iteration $K$ with accuracy $\varepsilon_K$, then with probability at least $1 - \delta$ it converges, with a number of samples upper bounded by:*

$$n \leq \tilde{\mathcal{O}}\left( \frac{4\gamma^2 R^2_{\max} SA}{(1-\gamma)^4 \varepsilon_K^2} \right). \tag{21}$$

*Proof.* Compare Lemma B.15 for the IRC solver with Lemma B.9 for the IRL solver. Using the proof techniques from (Metelli et al., 2021, Theorem 5.1), we derive the following sample complexity for the IRC solver. $\qquad\square$

*Remark* B.16. Compared to the IRL solver, the IRC solver utilizes the known reward signals for constraint inference. This method treats $r$ as "imperfect" rewards and learns a correction term $\Delta r(s, a)$ to incorporate constraint signals. Note that the IRC solver does not explicitly model the constraints during inference since the solver considers an unconstrained RL problem instead of CRL during policy updates.

## B.6 THEORETICAL RESULTS OF ICRL SOLVER

In this part, we provide additional results regarding the ICRL solver for constraint inference.

**Inverse Constrained Reinforcement Learning (ICRL).** ICRL solvers infer the constraint respected by the expert agents from their demonstrations. An ICRL solver admits the following assumptions.

**Assumption B.17.** *Either of the following two statements holds:*
*(i) The constraint in Eq. (PI) is a hard constraint such that $\epsilon = 0$;*
*(ii) The constraint in Eq. (PI) is a soft constraint such that $\epsilon > 0$, and the expert policy is deterministic.*

**Lemma 4.3.** *(Feasible cost set implicit (Yue et al., 2024, Lemma 4.3)). Under Assumption B.17, let $\mathfrak{P}$ be an ICI problem and $\mathbb{S}_{ICRL}$ be the ICRL solver, then $c$ is a feasible cost, i.e., $c \in \mathcal{C}_{\mathfrak{P}}$ if and only if $\forall(s, a) \in \mathcal{S} \times \mathcal{A}$:*

(i) if $\pi^E(a|s) > 0$, i.e., $(s, a)$ follows the expert policy:

$$Q_{\mathcal{M} \cup c}^{c, \pi^E}(s, a) - V_{\mathcal{M} \cup c}^{c, \pi^E}(s) = 0. \tag{22}$$

(ii) if $\pi^E(a|s) = 0$ and $A_{\mathcal{M} \cup c}^{r, \pi^E}(s, a) > 0$, i.e., $(s, a)$ violates the constraint:

$$Q_{\mathcal{M} \cup c}^{c, \pi^E}(s, a) - V_{\mathcal{M} \cup c}^{c, \pi^E}(s) > 0. \tag{23}$$

(iii) if $\pi^E(a|s) = 0$ and $A_{\mathcal{M} \cup c}^{r, \pi^E}(s, a) \leq 0$, i.e., $(s, a)$ is in the non-critical region:

$$Q_{\mathcal{M} \cup c}^{c, \pi^E}(s, a) - V_{\mathcal{M} \cup c}^{c, \pi^E}(s) \leq 0. \tag{24}$$

**Lemma B.18.** *Let $\mathfrak{P}$ be an ICI problem and $\mathbb{S}_{ICRL}$ be the ICRL solver. A Q-function satisfies the condition of Lemma 4.3 if and only if there exists $\zeta \in \mathbb{R}_{\geq 0}^{\mathcal{S} \times \mathcal{A}}$ and $V^c \in \mathbb{R}^{\mathcal{S}}$ such that:*

$$Q_{\mathcal{M} \cup c}^{c, \pi^E} = A_{\mathcal{M} \cup c}^{r, \pi^E} \zeta + EV^c, \tag{25}$$

*where the expansion operator $E$ satisfies $(Ef)(s) = f(s, a)$.*

**Lemma B.19.** *(Feasible cost set explicit (Yue et al., 2024, Lemma 4.4)). Let $\mathfrak{P}$ be an ICI problem and $\mathbb{S}_{ICRL}$ be the ICRL solver. $c$ is a feasible cost, i.e., $c \in \mathcal{C}_{\mathfrak{P}}$ if and only if there exist $\zeta \in \mathbb{R}_{\geq 0}^{\mathcal{S} \times \mathcal{A}}$ and $V^c \in \mathbb{R}^{\mathcal{S}}$ such that:*

$$c = A_{\mathcal{M} \cup c}^{r, \pi^E} \zeta + (E - \gamma P_{\mathcal{T}}) V^c. \tag{26}$$

Similarly, we show how the estimation error on the environmental dynamic and on the expert policy propagates to the cost function.

**Lemma B.20.** *(Error Propagation (Yue et al., 2024, Lemma 4.5)). Let $\mathfrak{P} = (\mathcal{M}, \pi^E)$ and $\widehat{\mathfrak{P}} = (\widehat{\mathcal{M}}, \widehat{\pi}^E)$ be two ICI problems. For any $c \in \mathcal{C}_{\mathfrak{P}}$ satisfying $c = A_{\mathcal{M} \cup c}^{r, \pi^E} \zeta + (E - \gamma P_{\mathcal{T}}) V^c$ and $\|c\|_{\infty} \leq C_{\max}$, there exists $\widehat{c} \in \mathcal{C}_{\widehat{\mathfrak{P}}}$ such that element-wise it holds that:*

$$|c - \widehat{c}| \leq \gamma \left| (P_{\mathcal{T}} - \widehat{P_{\mathcal{T}}}) V^c \right| + \left| A_{\mathcal{M} \cup c}^{r, \pi^E} - A_{\widehat{\mathcal{M}} \cup r}^{r, \widehat{\pi}^E} \right| \zeta. \tag{27}$$

*Furthermore, $\|V^c(s)\|_{\infty} \leq C_{\max}/(1 - \gamma)$ and $\|\zeta\|_{\infty} \leq C_{\max}/ \min_{(s,a)}^+ |A_{\mathcal{M} \cup c}^{r, \pi^E}|$.*

**Lemma B.21.** *(Yue et al., 2024, Lemma 4.6) For a given policy $\pi$, let $A_{\mathcal{M} \cup r}^{r, \pi}$ denote the reward advantage function based on the original MDP $\mathcal{M}$. For an estimated policy $\widehat{\pi}$, let $A_{\widehat{\mathcal{M}} \cup c}^{r, \widehat{\pi}}$ denote the reward advantage function based on the estimated MDP $\widehat{\mathcal{M}}$. Then, we have*

$$\left| A_{\mathcal{M} \cup c}^{r, \pi} - A_{\widehat{\mathcal{M}} \cup c}^{r, \widehat{\pi}} \right| \leq \frac{2\gamma}{1 - \gamma} \left| (\widehat{P_{\mathcal{T}}} - P_{\mathcal{T}}) V_{\widehat{\mathcal{M}}}^{r, \widehat{\pi}} \right| + \frac{\gamma(1 + \gamma)}{1 - \gamma} \left| (\pi - \widehat{\pi}) P_{\mathcal{T}} V_{\mathcal{M}}^{r, \pi} \right|. \tag{28}$$

**Theorem 4.5.** *(Sample Complexity of ICRL solver(Yue et al., 2024, Theorem C.9)). If an ICRL solver terminates at iteration $K$ with the updated accuracy $\varepsilon_K$, then with probability at least $1 - \delta$, it converges with a number of samples upper bounded by*

$$n \leq \widetilde{\mathcal{O}} \left( \frac{\gamma^2 \sigma^2 SA}{(1 - \gamma)^6 \varepsilon_K^2} \right). \tag{29}$$

*where $\sigma = \sqrt{3} \gamma C_{\max} \left( R_{\max}(3 + \gamma)/ \min^+ \left| A_{\mathcal{M} \cup c}^{r, \pi^E} \right| + (1 - \gamma) \right)$.*

*Remark B.22.* Compared with the IRC solver, the ICRL solver considers a CMDP environment instead of an MDP one, so it explicitly models constraints during policy updates.

## B.7 NUMERICAL ANALYSIS OF THE EXAMPLE IN FIGURE 1

As illustrated in Figure 1, the basic environment contains $2 \times 5$ grids, where the agent navigates from the starting location $(0, 0)$ (the left bottom corner) to the target location $(0, 4)$ (the left top corner) with four possible actions, i.e., Up, Down, Left and Right. In the source environment, $(0, 4)$ (colored blue) is assigned with a reward of 10, $s^c = (0, 2)$ (colored orange) contains a hard constraint that

must not be accessed, while all other locations are assigned with 0 rewards and 0 costs. The discount factor $\gamma = 0.7$ and there is no stochasticity in the source environment. For better readability, in this part, we simplify $Q_{\mathcal{M}\cup(r/r'+\Delta r)}^{r/r'+\Delta r,\pi^E}(s,a)$ as $Q^{r/r'+\Delta r}(a)$ and $V_{\mathcal{M}\cup(r/r'+\Delta r)}^{r/r'+\Delta r,\pi^E}(s,a)$ as $V^{r/r'+\Delta r}(a)$.

If we do not add the reward correction term $\Delta r$, under the current reward function $r((0,4)) = 10$, the agent chooses to go Up at $(0,1)$, which violates the hard constraint and deviates from the expert policy. To guide the agent to go right at $(0,1)$ instead of going up, a feasible reward correction term is $\Delta r((0,2)) = -3$ and are zero at all the other locations, which, together with reward function $r((0,4)) = 10$, leads to

$$Q^{r+\Delta r,\pi^E}((0,1),\text{Up}) = \Delta r((0,2)) + \gamma \times V^{r+\Delta r,\pi^E}((0,2)) = -3 + 0.7 \times 7 = 1.9,$$

$$Q^{r+\Delta r,\pi^E}((0,1),\text{Right}) = \Delta r((1,1)) + \gamma \times V^{r+\Delta r,\pi^E}((1,1)) = 0 + 0.7 \times 3.43 = 2.401.$$

This ensures the optimality of the expert policy at state $(0,1)$ since

$$Q^{r+\Delta r,\pi^E}((0,1),\text{Up}) < Q^{r+\Delta r,\pi^E}((0,1),\text{Right}),$$

meaning the safe and optimal action Right is chosen. Figure 5 illustrates the source environment, the optimal value function regarding $r + \Delta r$, and the optimal policy at each state.

Suppose we have a target environment, with $r'((0,4)) = 20$, and stochasticity of 0.3 to state $(1,4),(1,3),(1,1)$ and $(0,1)$ of the environment and there remains no stochasticity to all the other states. Figure 6 (middle) shows the new reward value function regarding $r' + \Delta r$. Figure 6 (right) shows the policy generated by $r'$ with $\Delta r((0,2)) = -3$. We observe that $\Delta r((0,2)) = -3$ is not enough to penalize the going up action at $(0,1)$, since

$$Q^{r'+\Delta r,\pi^E}((0,1),\text{Up}) = \Delta r((0,2)) + \gamma \times V^{r'+\Delta r,\pi^E}((0,2)) = -3 + 0.7 \times 14 = 6.8,$$

$$Q^{r'+\Delta r,\pi^E}((0,1),\text{Right}) = \Delta r((1,1)) + \gamma \times V^{r'+\Delta r,\pi^E}((1,1)) = 0 + 0.7 \times 5.78 = 4.046,$$

$$Q^{r'+\Delta r,\pi^E}((0,1),\text{Up}) > Q^{r'+\Delta r,\pi^E}((0,1),\text{Right}),$$

meaning the constraint-violating action Up is chosen.

Next, we validate Theorem 5.3. At state $(0,1)$, $a^E = \text{Right}$, $a^C = \text{Up}$, $a_1^O = \text{Down}$, $a_2^O = \text{Left}$. Through numerical calculation, we derive the following results, which match the results of Theorem 5.3.

- $Q^{r+\Delta r}(a^E) - Q^{r+\Delta r}(a^C) = 0.501 \geq 0$;

- $Q^{r+\Delta r}(a^E) - Q^{r+\Delta r}(a_1^O) = 1.225 \geq 0$; $Q^{r+\Delta r}(a^E) - Q^{r+\Delta r}(a_2^O) = 0.720 \geq 0$;

- $Q^{r+\Delta r}(a^E) - Q^{r+\Delta r}(a^C) = 0.501 < (Y')^{-1}(Y - Y')Q^{r+\Delta r}[a^C - a^E] = 3.701$;

- $Q^{r+\Delta r}(a_1^O) - Q^{r+\Delta r}(a^C) = -0.724 < (Y')^{-1}(Y - Y')Q^{r+\Delta r}[a^C - a_1^O] = 4.312$;

$$Q^{r+\Delta r}(a_2^O) - Q^{r+\Delta r}(a^C) = -0.219 < (Y')^{-1}(Y - Y')Q^{r+\Delta r}[a^C - a_2^O] = 4.060.$$

## B.8 THEORETICAL RESULTS OF TRANSFERABILITY IN SAFETY

### B.8.1 THE HARD CONSTRAINT SCENARIO

**Lemma 5.2.** *Suppose a hard constraint scenario. For any $(s',a') \in \mathcal{G}$, the feasible cost function $\hat{c}$ inferred by the ICRL solver (3.1) can prevent the visitation to $(s',a')$ in the target CMDP.*

*Proof.* For any $(s',a') \in \mathcal{G}$ and efficiently small cost estimation error $\varepsilon$, there must be $\hat{c}(s',a') > 0$ by the ICRL solver, which bans action $a'$ at state $s'$. In a hard constraint scenario, when $\hat{c}$ is transferred to a target CMDP, this $\hat{c}(s',a')$ will also ban this action $a'$ at state $s'$, since any policy with $\pi(a'|s') > 0$ leads to a violation of the hard constraint. □

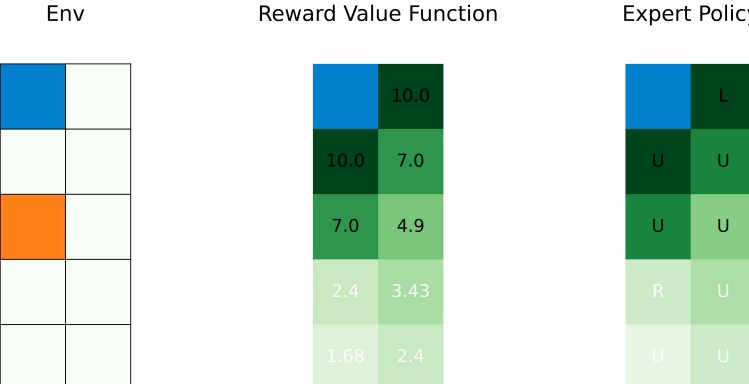

Figure 5: The basic environment (left), the source reward value function under $r$ and $P_{\mathcal{T}}$ (middle), and the expert policy of the example in Figure 1 (right).

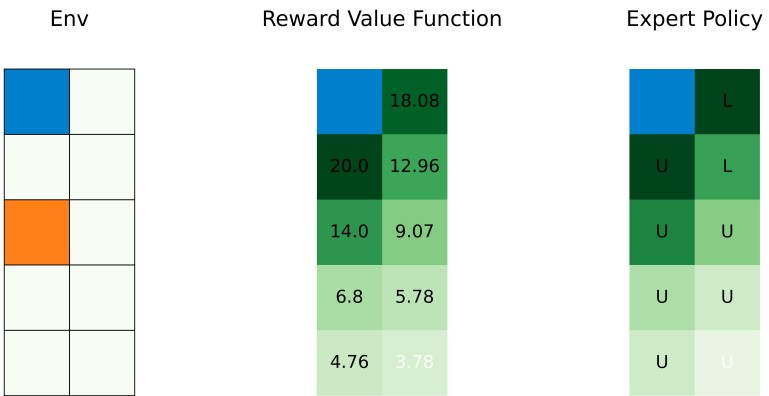

Figure 6: The basic environment (left), the target reward value function under $r'$ and $P'_{\mathcal{T}}$ (middle), and the generated policy (right).

**Theorem 5.3.** *Suppose a hard constraint scenario. At state $s$, let $a^E$ denote the expert action , $a^C$ denote the action that satisfies $(s, a^C) \in \mathcal{G}$ and $a^O$ denote the other actions. $\forall r' \in [0, R_{\max}]^{\mathcal{S} \times \mathcal{A}}$ and $\forall P'_{\mathcal{T}} \in \Delta^{\mathcal{S}}_{\mathcal{S} \times \mathcal{A}}$, if $\exists s \in \mathcal{S}, \forall a^E, a^O \in \mathcal{A}, \exists a^C \in \mathcal{A}$ that satisfies the following condition, then the reward correction term $\Delta r$ constructed by such Q-functions violates safety in the target CMDP,*

$$Q^{r+\Delta r, \pi^E}_{\mathcal{M} \cup (r + \Delta r)}(\mathbf{e}_{(s, a^E)} - \mathbf{e}_{(s, a^C)}) \geq 0, Q^{r+\Delta r, \pi^E}_{\mathcal{M} \cup (r + \Delta r)}(\mathbf{e}_{(s, a^E)} - \mathbf{e}_{(s, a^O)}) \geq 0,$$

$$Q^{r+\Delta r, \pi^E}_{\mathcal{M} \cup (r + \Delta r)}(\mathbf{e}_{(s, a^E)} - \mathbf{e}_{(s, a^C)}) < (Y')^{-1} \left[ r' - r + (Y - Y') Q^{r+\Delta r, \pi^E}_{\mathcal{M} \cup (r + \Delta r)} \right] (\mathbf{e}_{(s, a^C)} - \mathbf{e}_{(s, a^E)}),$$

$$Q^{r+\Delta r, \pi^E}_{\mathcal{M} \cup (r + \Delta r)}(\mathbf{e}_{(s, a^O)} - \mathbf{e}_{(s, a^C)}) < (Y')^{-1} \left[ r' - r + (Y - Y') Q^{r+\Delta r, \pi^E}_{\mathcal{M} \cup (r + \Delta r)} \right] (\mathbf{e}_{(s, a^C)} - \mathbf{e}_{(s, a^O)}),$$

*where $Y = (I_{\mathcal{S} \times \mathcal{A}} - \gamma P_{\mathcal{T}} \pi^E)$, $Y' = (I_{\mathcal{S} \times \mathcal{A}} - \gamma P'_{\mathcal{T}} (\pi')^E)$ and $\mathbf{e}_{(s,a)}$ denotes a vector with value of 1 at the index $(s, a)$ and 0 elsewhere.*

*Proof.* For better readability, in this part, we denote $Q^{r'} = Q^{r' + \Delta r, (\pi')^E}_{\mathcal{M}' \cup (r' + \Delta r)}$ and $Q^r = Q^{r + \Delta r, \pi^E}_{\mathcal{M} \cup (r + \Delta r)}$. Since, at each state $s \in \mathcal{S}$, the agent chooses its action based on the maximum Q-function, we study the Q-function in the source and target CMDP.

$$Q^r(s, a) = (I_{\mathcal{S} \times \mathcal{A}} - \gamma P_{\mathcal{T}} \pi^E)^{-1} r^c(s, a) = (I_{\mathcal{S} \times \mathcal{A}} - \gamma P_{\mathcal{T}} \pi^E)^{-1} [r(s, a) + \Delta r(s, a)]$$

$$Q^{r'}(s,a) = (I_{\mathcal{S}\times\mathcal{A}} - \gamma P'_{\mathcal{T}}(\pi')^E)^{-1}(r^c)'(s,a) = (I_{\mathcal{S}\times\mathcal{A}} - \gamma P'_{\mathcal{T}}(\pi')^E)^{-1}[r'(s,a) + \Delta r(s,a)]$$

We next show how the Q-function in the target CMDP is shifted from the source CMDP.

$$Q^r(s,a) = (I_{\mathcal{S}\times\mathcal{A}} - \gamma P_{\mathcal{T}}\pi^E)^{-1}[r + \Delta r](s,a) \tag{30}$$

$$\begin{aligned}
Q^{r'}(s,a) &= (I_{\mathcal{S}\times\mathcal{A}} - \gamma P'_{\mathcal{T}}(\pi')^E)^{-1}[r' + \Delta r](s,a) \\
&= (I_{\mathcal{S}\times\mathcal{A}} - \gamma P'_{\mathcal{T}}(\pi')^E)^{-1}[r' - r + (I_{\mathcal{S}\times\mathcal{A}} - \gamma P_{\mathcal{T}}\pi^E)Q^r](s,a) \\
&= (Y')^{-1}[(r' - r) + YQ^r](s,a) \\
&= (Y')^{-1}[(r' - r) + (Y - Y' + Y')Q^r](s,a) \\
&= (Y')^{-1}[(r' - r) + (Y - Y')Q^r](s,a) + Q^r(s,a). \tag{31}
\end{aligned}$$

Based on Eq. (31), we have

$$\begin{aligned}
Q^{r'}(s,a^E) &= (Y')^{-1}[r' - r + (Y - Y')Q^r](s,a^E) + Q^r(s,a^E) \\
Q^{r'}(s,a^C) &= (Y')^{-1}[r' - r + (Y - Y')Q^r](s,a^C) + Q^r(s,a^C) \\
Q^{r'}(s,a^O) &= (Y')^{-1}[r' - r + (Y - Y')Q^r](s,a^O) + Q^r(s,a^O). \tag{32}
\end{aligned}$$

The first two inequalities result from Lemma 4.1, aiming to ensure the optimality of expert policy. The third and fourth inequalities are derived by substituting Eqs. (32) into $Q^{r'}(s,a^E) < Q^{r'}(s,a^C)$ and $Q^{r'}(s,a^O) < Q^{r'}(s,a^C)$ to construct target Q-functions that drive the agent choose constraint-violating actions $a^C$.

Given $Y = (I_{\mathcal{S}\times\mathcal{A}} - \gamma P_{\mathcal{T}}\pi^E)$ and $Y' = (I_{\mathcal{S}\times\mathcal{A}} - \gamma P'_{\mathcal{T}}(\pi')^E)$, we have

$$\begin{aligned}
Y - Y' &= \gamma(P'_{\mathcal{T}}(\pi')^E - P_{\mathcal{T}}\pi^E) \\
&= \gamma[P'_{\mathcal{T}}(\pi')^E - P_{\mathcal{T}}(\pi')^E + P_{\mathcal{T}}(\pi')^E - P_{\mathcal{T}}\pi^E] \\
&= \underbrace{\gamma[(P'_{\mathcal{T}} - P_{\mathcal{T}})(\pi')^E]}_{\text{A: transition difference term}} + \underbrace{\gamma[P_{\mathcal{T}}((\pi')^E - \pi^E)]}_{\text{B: expert policy difference term}} \tag{33} \\
&= A + B
\end{aligned}$$

Based on (33), we can further split the increment from $Q^r(s,a)$ to $Q^{r'}(s,a)$ in (31) into:

$$Q^{r'}(s,a) - Q^r(s,a) = \left[ \underbrace{(Y')^{-1}(r' - r)}_{\text{reward transfer shift}} + \underbrace{(Y')^{-1}AQ^r}_{\text{transition transfer shift}} + \underbrace{(Y')^{-1}BQ^r}_{\text{expert policy transfer shift}} \right](s,a). \tag{34}$$

$\square$

We observe how transfer from the source CMDP to the target CMDP affects the Q-value function, driven by variations in the reward functions, transition dynamics, and the expert policy.

### B.8.2 THE SOFT CONSTRAINT SCENARIO

In this part, we extend our theory from the hard constraint scenario to the soft constraint scenario. Unlike the hard constraint scenario, where we construct a constraint set on state-action pairs (i.e., $\{(s,a)|c(s,a) > 0\}$) that a sensible agent should never visit, the soft constraint scenario is more challenging. Given a threshold $\epsilon > 0$, no matter how large a cost $c(s,a)$ is, the agent can always visit $(s,a)$ after a sufficiently large number of steps as the effective cost of visiting $(s,a)$ can be reduced below the threshold $\epsilon$ via the discount factor $\gamma < 1$ raised to the power of $\lfloor \log_{c(s,a)} \epsilon \rfloor + 1$.

For IRC solvers, in the soft constraint scenario, safety is also not guaranteed by the transferred reward correction term in the target environment. Lemma 5.3 still applies, as long as $\mathcal{G} \neq \varnothing$.

For ICRL solvers, the key distinction between the hard and soft constraint scenarios is that the inferred cost function $c$ no longer guarantees safety, since $Q^{c,(\pi')^E}_{\mathcal{M}'\cup c}(s,a) - V^{c,(\pi')^E}_{\mathcal{M}'\cup c}(s) > 0$ (case (ii) in Lemma 4.3) is not necessarily satisfied for $(s,a) \in \mathcal{G}$. However, we demonstrate that the

ICRL solver still offers better transferability than the IRC solver in safety, as it remains unaffected by reward transfer shifts.

Since at each state $s \in \mathcal{S}$, the agent examines whether an action is safe or not based on the cost Q-function, we first study the cost Q-function in the source and target CMDP, which are defined as:

$$Q^c(s,a) = Q_{\mathcal{M} \cup c}^{c,(\pi)^E} = \left[(I_{\mathcal{S} \times \mathcal{A}} - \gamma P_{\mathcal{T}} \pi^E)^{-1} c\right](s,a) \tag{35}$$

$$Q^{c'}(s,a) = Q_{\mathcal{M}' \cup c}^{c,(\pi')^E} = \left[(I_{\mathcal{S} \times \mathcal{A}} - \gamma P_{\mathcal{T}}'(\pi')^E)^{-1} c\right](s,a) \tag{36}$$

Similar to the transferability of reward correction terms, we further discuss the influence of different reward functions, transition models, and expert policies on the transferability of cost functions. We next show how the cost Q-function in the target CMDP is shifted from the source CMDP.

From (35) and (36), we obtain

$$\begin{aligned}
Q^{c'}(s,a) &= [(Y')^{-1} Y Q^c](s,a) \\
&= [(Y')^{-1}(Y - Y' + Y')Q^c](s,a) \\
&= [(Y')^{-1}(Y - Y')Q^c](s,a) + Q^c(s,a)
\end{aligned} \tag{37}$$

Given (33), the increment in cost Q-functions can be split into:

$$Q^{c'}(s,a) - Q^c(s,a) = \left[ \underbrace{(Y')^{-1} A Q^c}_{\text{transition transfer shift}} + \underbrace{(Y')^{-1} B Q^c}_{\text{expert policy transfer shift}} \right](s,a) \tag{38}$$

By comparing (34) with (38), we can clearly the differences in transferring reward correction terms and cost functions become evident. The transfer of cost functions by the ICRL solver is influenced by: (1) transition shifts and (2) expert policy shifts. In contrast, the transfer of reward correction terms by the IRC solver is affected by: (1) reward shifts, (2) transition shifts, and (3) expert policy shifts.

Next, we identify the condition under which a cost function $c$ is not transferable. At state $s$, for $a^C \in \mathcal{G}$, we further distinguish two cases. Let $a^{C_1}$ denote the action that satisfies $(s, a^{C_1}) \in \{(s,a)|A_{\mathcal{M} \cup c^E}^{r, \pi^E}(s,a) > 0\} \cap \{(s,a)|A_{\mathcal{M}' \cup (c')^E}^{r',(\pi')^E}(s,a) > 0\}$ and $a^{C_2}$ denote the action that satisfies $(s, a^{C_2}) \in \{(s,a)|A_{\mathcal{M} \cup c^E}^{r, \pi^E}(s,a) > 0\} \backslash \{(s,a)|A_{\mathcal{M}' \cup (c')^E}^{r',(\pi')^E}(s,a) > 0\}$.

Based on Eq. (37), we have

$$\begin{aligned}
Q^{c'}(s, a^E) &= [(Y')^{-1}(Y - Y')Q^c](s, a^E) + Q^c(s, a^E) \\
Q^{c'}(s, a^O) &= [Y')^{-1}(Y - Y')Q^c](s, a^O) + Q^c(s, a^O) \\
Q^{c'}(s, a^{C_1}) &= [(Y')^{-1}(Y - Y')Q^c](s, a^{C_1}) + Q^c(s, a^{C_1}) \\
Q^{c'}(s, a^{C_2}) &= [(Y')^{-1}(Y - Y')Q^c](s, a^{C_2}) + Q^c(s, a^{C_2}) \\
Q^{c'}(s, (a')^E) &= [(Y')^{-1}(Y - Y')Q^c](s, (a')^E) + Q^c(s, (a')^E)
\end{aligned} \tag{39}$$

A sufficient condition that the ICRL solver fails would be choosing $a^{C_1}$ in the target CMDP. Specifically, we should show the condition under which with new reward function $r'$ and new transition $P_{\mathcal{T}}'$, $\exists s \in \mathcal{S}, \forall a^E \in \mathcal{A}$ and $\forall a^O \in \mathcal{A}, \exists a^{C_1} \in \mathcal{A}: Q^{c'}(s, (a')^E) > Q^{c'}(s, a^{C_1})$ ($a^{C_1}$ does not violate the constraint but achieves higher rewards). We formally state this condition in the following lemma.

**Lemma B.23.** *$\forall r' \in [0, R_{\max}]^{\mathcal{S} \times \mathcal{A}}$ and $\forall P_{\mathcal{T}}' \in \Delta_{\mathcal{S} \times \mathcal{A}}^{\mathcal{S}}$, if $\exists s \in \mathcal{S}, \forall a^E \in \mathcal{A}$ and $\forall a^O \in \mathcal{A}$, $\exists a^{C_1} \in \mathcal{A}$ that satisfies the following condition, then a cost function $c$ constructed by such cost Q-functions is non-transferable in the target CMDP,*

$$\begin{aligned}
& Q^c(s, a^E) < Q^c(s, a^{C_1}), Q^c(s, a^E) < Q^c(s, a^{C_2}), Q^c(s, a^E) \geq Q^c(s, a^O) \\
& Q^c(s, (a')^E) - Q^c(s, a^{C_1}) > [(Y')^{-1}(Y - Y')Q^c](\mathbf{e}_{(s, a^{C_1})} - \mathbf{e}_{(s, (a')^E)})
\end{aligned} \tag{40}$$

*where $\mathbf{e}_{(s,a)}$ denotes a vector with value of 1 at the index $(s,a)$ and 0 elsewhere.*

*Proof.* The first three inequalities result from Lemma 4.3, aiming to ensure the optimality of expert policy in the source environment. For the fourth inequality, based on Eqs. (39), the condition

$$Q^{c'}(s, (a')^E) > Q^{c'}(s, a^{C_1})$$

is equivalent to

$$[(Y')^{-1}(Y - Y')Q^c](s, (a')^E) + Q^c(s, (a')^E) > [(Y')^{-1}(Y - Y')Q^c](s, a^{C_1}) + Q^c(s, a^{C_1})$$

. By simple transpositions, we obtain

$$Q^c(s, (a')^E) - Q^c(s, a^{C_1}) > [(Y')^{-1}(Y - Y')Q^c](s, a^{C_1}) - [(Y')^{-1}(Y - Y')Q^c](s, (a')^E)$$
$$= [(Y')^{-1}(Y - Y')Q^c](\mathbf{e}_{(s, a^{C_1})} - \mathbf{e}_{(s, (a')^E)}). \tag{41}$$

$\square$

**Corollary B.24.** *If two additional assumptions are satisfied: 1) the transition model does not change, i.e., $P'_{\mathcal{T}} = P_{\mathcal{T}}$; 2) the optimal policy of source CMDP is the optimal policy of target CMDP, i.e., $\Pi^*_{\mathcal{M}_c} \subseteq \Pi^*_{\mathcal{M}'_c}$, then a feasible cost function $\hat{c}$ inferred by the ICRL solver (3.1) can prevent any visitation to $(s', a') \in \mathcal{G} \neq \varnothing$ in the target CMDP.*

*Proof.* Suppose $(s', a') \in \mathcal{G} \neq \varnothing$, at state $s'$, since $a'$ can bring more rewards both in the source CMDP and in the target CMDP, $a'$ should be abandoned in both CMDPs. This means, in a soft constraint scenario, the expert policy at state $s'$ must reach the threshold (Yue et al., 2024, Lemma 4.2). Since the constraint condition in case (ii) of Lemma 4.3 should be satisfied both in the source and target CMDP, for $\forall \theta \in (0, 1]$, policy $\pi'(a|s') = \begin{cases} \theta & , a = a' \\ 1 - \theta, a \sim \pi^E = (\pi')^E \end{cases}$ violates the soft constraint in target CMDP. As a result, action $a'$ is abandoned in the target CMDP after transferring. $\square$

### B.9 THEORETICAL RESULTS OF TRANSFERABILITY IN OPTIMALITY

**Proposition B.25.** *Let $H_\gamma := 1/(1 - \gamma)$, $R := \max_{r \in \mathcal{R}} \|\check{r}\|_\infty$, and $D = \max_{\check{r}, r' \in \mathcal{R}} \|r - r'\|_2$. Suppose that $\alpha < 2D$, then for the Shannon entropy regularization, $\eta$ and $\sigma_{\mathcal{R}}$ can be obtained from (Schlaginhaufen & Kamgarpour, 2024, Proposition D.8):*

$$\eta = \alpha \mu_{min}/H_\gamma^2 \quad and \quad \sigma_{\mathcal{R}} = \frac{\mu_{min}\left(1 - \frac{\alpha}{2D}\right)\exp\left(\frac{-2RH_\gamma}{\alpha}\right)}{D|\mathcal{S}||\mathcal{A}|^{2+H_\gamma}}. \tag{42}$$

**Lemma B.26.** *Suppose Assumptions 5.4 hold, and let $r, r' \in \mathcal{R}$. Then, we have*

$$\frac{\sigma_{\mathcal{R}}}{2}\|[r]_{\mathcal{U}} - [r']_{\mathcal{U}}\|_2^2 \leq \ell(r', \mathsf{RL}(r)) = D_h\left(\mathsf{RL}(r), \mathsf{RL}(r')\right) \leq \frac{1}{2\eta}\|[r]_{\mathcal{U}} - [r']_{\mathcal{U}}\|_2^2,$$

*for some problem-dependent constant $\sigma_{\mathcal{R}} > 0$.*

*Proof.* This lemma directly comes from (Schlaginhaufen & Kamgarpour, 2024, Lemma 3.5). $\square$

**Lemma B.27.** *Consider $x, y \in \mathbb{R}^n$ and two subspaces $\mathcal{V}, \mathcal{W} \subset \mathbb{R}^n$ of dimension $m < n$. Then,*

$$\|[x]_{\mathcal{W}} - [y]_{\mathcal{W}}\|_2 \leq \|\Pi_{\mathcal{W}} - \Pi_{\mathcal{W}}\| \cdot \|x - y\|_2 + \|[x]_{\mathcal{V}} - [y]_{\mathcal{V}}\|_2,$$

*where $\|\Pi_{\mathcal{W}} - \Pi_{\mathcal{V}}\| = \sin(\theta_{\max}(\mathcal{V}, \mathcal{W}))$.*

*Proof.* This lemma directly comes from (Schlaginhaufen & Kamgarpour, 2024, Proposition D.10). $\square$

**Definition B.28.** *(Cost equivalence).* We extend the results of reward equivalence (feasible reward set) in IRL settings (Schlaginhaufen & Kamgarpour, 2024) to cost equivalence (feasible cost set) in ICRL settings. Given a linear subspace $\mathcal{V} \subseteq \mathbb{R}^{\mathcal{S} \times \mathcal{A}}$, the quotient space $\mathbb{R}^{\mathcal{S} \times \mathcal{A}}/\mathcal{V}$ is the set of all equivalence classes $[r - \lambda^* c]_{\mathcal{V}} := \{r - \lambda^* c' \in \mathbb{R}^{\mathcal{S} \times \mathcal{A}} : \lambda^*(c' - c) \in \mathcal{V}\}$. On quotient space $\mathbb{R}^{\mathcal{S} \times \mathcal{A}}/\mathcal{V}$, we define the quotient norm $\|[x]_{\mathcal{V}}\|_2 := \min_{v \in \mathcal{V}}\|x + v\|_2 = \|\Pi_{\mathcal{V}^\perp}x\|_2$ and we say that $c$ and $\hat{c}$ are close in $\mathbb{R}^{\mathcal{S} \times \mathcal{A}}/\mathcal{V}$ given $\lambda^*$, if $\|[r - \lambda^* c]_{\mathcal{V}} - [r - \lambda^* c']_{\mathcal{V}}\|_2 = \|[(r - \lambda^* c) - (r - \lambda^* c')]_{\mathcal{V}}\|_2 = \|[\lambda^*(c' - c)]_{\mathcal{V}}\|_2$

is small. The expert's cost is said to be recovered by $\hat{c}$ up to $[\cdot]_{\mathcal{V}}$ if $r - \lambda^* \hat{c} \subseteq [r - \lambda^* c^E]_{\mathcal{V}}$. In this paper, we consider the equivalence relations induced by the subspace of potential shaping transformations (Ng et al., 1999), i.e., $\mathcal{V} = \mathcal{U}_{P_{\mathcal{T}}} := \text{im}(E - \gamma P_{\mathcal{T}})$. Revisiting Eq. (26), this subspace is equivalent to the way we construct a feasible cost set.

**Theorem 5.7.** *Let $P'_{\mathcal{T}}$ be the target transition law, the source and target ground-truth constraints are the same $(c')^E = c^E$ and $d_1 = \|[c^E - \hat{c}]_{\mathcal{U}_{P'_{\mathcal{T}}}}\|_2$. Suppose that Assumption 5.4 holds. If $\ell^{r', (\lambda')^*}_{P_{\mathcal{T}}}(\hat{c}, \text{CRL}(c^E)) \leq \varepsilon_1$, then $\hat{c}$ is $\varepsilon$-transferable to the target environment with*

$$\varepsilon = 2 \max \left\{ d_1^2 \sin \left( \theta_{\max}(P_{\mathcal{T}}', P_{\mathcal{T}}) \right)^2 / 2, 2\varepsilon_1 / \sigma_{\mathcal{R}} \right\} / \eta, \tag{43}$$

*where $\sigma_{\mathcal{R}}$ and $\eta$ are regularity constants, given in Appendix B.25.*

*Proof.* For better readability, we denote $(\hat{r})' = r' - (\lambda')^* \hat{c}$ and $(\check{r}')^E = r' - (\lambda')^* c^E$. It follows from Lemma B.26 that $\left\| [(\check{r}')^E]_{\mathcal{U}_{P_{\mathcal{T}}}} - [(\hat{r})']_{\mathcal{U}_{P_{\mathcal{T}}}} \right\|_2 \leq \sqrt{2\varepsilon_1 / \sigma_{\mathcal{R}}}$. By choosing the closest $(\check{r}')^E$ and $(\hat{r})'$ in subspace $\mathcal{U}_{P'_{\mathcal{T}}}$ in Lemma B.27, we then have that

$$\left\| [(\check{r}')^E]_{\mathcal{U}_{P'_{\mathcal{T}}}} - [(\hat{r})']_{\mathcal{U}_{P'_{\mathcal{T}}}} \right\|_2 \leq \sin \left( \theta_{\max}(P'_{\mathcal{T}}, P_{\mathcal{T}}) \right) \left\| [(\check{r}')^E - (\hat{r})']_{\mathcal{U}_{P'_{\mathcal{T}}}} \right\|_2 + \left\| [(\check{r}')^E]_{\mathcal{U}_{P_{\mathcal{T}}}} - [(\hat{r})']_{\mathcal{U}_{P_{\mathcal{T}}}} \right\|_2$$

$$\leq d_1 \sin \left( \theta_{\max}(P'_{\mathcal{T}}, P_{\mathcal{T}}) \right) + \sqrt{2\varepsilon_1 / \sigma_{\mathcal{R}}} \tag{44}$$

Hence, applying Lemma B.26 again yields

$$\ell^{r', (\lambda')^*}_{P'_{\mathcal{T}}} \left( \hat{c}, \text{CRL}_{P'_{\mathcal{T}}}((c')^E) \right) = \ell^{r', (\lambda')^*}_{P'_{\mathcal{T}}} \left( \hat{c}, \text{CRL}_{P'_{\mathcal{T}}}(c^E) \right)$$

$$\leq \frac{1}{2\eta} \left\| [(\check{r}')^E]_{\mathcal{U}_{P'_{\mathcal{T}}}} - [(\hat{r})']_{\mathcal{U}_{P'_{\mathcal{T}}}} \right\|_2^2$$

$$\leq \frac{(d_1 \sin \left( \theta_{\max}(P_{\mathcal{T}}', P_{\mathcal{T}}) \right) + \sqrt{2\varepsilon_1 / \sigma_{\mathcal{R}}})^2}{2\eta}$$

$$\leq \frac{2 \max \left\{ d_1^2 \sin \left( \theta_{\max}(P_{\mathcal{T}}', P_{\mathcal{T}}) \right)^2 / 2, 2\varepsilon_1 / \sigma_{\mathcal{R}} \right\}}{\eta}. \tag{45}$$

$\square$

## C  COMPARISON IN SOFT CONSTRAINT SCENARIOS

We evaluate the performance of the ICRL solver in soft constraint scenarios in two aspects. In aspect one, only reward functions are different between source and target. In aspect two, only transition dynamics are different between source and target.

*Aspect One: In soft constraint scenarios, the ICRL solver still outperforms the IRC solver in the sense that it resists the variation between source and target reward functions.* Table 2 illustrates whether the inferred reward correction terms by the IRC solver violate the constraint in the target environment (safe ✓ or not ✗). We can see that with the increase in reward functions, the inferred reward correction terms tend to become unsafe while the inferred cost functions by ICRL solvers are safe still. Threshold $\epsilon = 0.015$ and ground truth costs are 1.

Table 2: Safety (safe ✓ or not ✗) of inferred reward correction terms by the IRC solver under different rewards in the target environments of Gridworld-1. T represents the terminal location (6,6) in Gridworld-1.

| different $r'(T) \uparrow$ | $r'(T) = 1$ | $r'(T) = 1.2$ | $r'(T) = 1.4$ | $r'(T) = 1.6$ | $r'(T) = 1.8$ | $r'(T) = 2$ |
|---|---|---|---|---|---|---|
| IRC solver | ✓ | ✓ | ✓ | ✗ | ✗ | ✗ |
| ICRL solver | ✓ | ✓ | ✓ | ✓ | ✓ | ✓ |

*Aspect Two: In soft constraint scenarios, ICRL solver can violate constraints due to compensated penalization by new transition dynamics.* We provide a case study to help the readers better understand this aspect. Reconsider the example in Figure 1 in a soft constraint scenario. We only set $c((0,2)) = 1$ and $\epsilon = 0.8 \times (0.7)^{-2} > 0$. Suppose in the source environment, there is no randomness for any chosen actions. The true cost function the expert follows is $c^E((0,2)) = 1$. In this case, the shortest path, i.e., going straight upward from (0,0) to (0,5), is forbidden by the expert because the path induces a cumulative cost of $1 \times (0.7)^{-2} > \epsilon$. One choice of the feasible cost function is $c((0,2)) = 0.85$, since $0.85 \times (0.7)^{-2} > \epsilon$ is sufficient to ban the shortest path. Now we transfer $c((0,2)) = 0.85$ from the source to the target environment. Suppose the target environment only differs from the source environment in the transition model. In the target environment, if the location (0,1) alters its transition model to be $P_{\mathcal{T}}((0,2)|(0,1),\text{Up}) = 0.9$ and $P_{\mathcal{T}}((1,1)|(0,1),\text{Up}) = 0.1$. The inferred optimal policy based on $c((0,2)) = 0.85$ visits $(0,2)$ because the shortest path has a cumulative cost of $0.85 \times 0.9 \times (0.7)^{-2} = 0.765 \times (0.7)^{-2} < \epsilon$ but should be forbidden because the policy of going straight upwards in the left column has a cumulative ground truth cost of $0.9 \times (0.7)^{-2} > \epsilon$.

# D    COMPARISON IN CONTINUOUS ENVIRONMENTS

For continuous environments, we apply an offline setting to compare the transferability of constraint knowledge inferred by IRC or ICRL solvers. The offline setting is different from the online setting in discrete environments. The expert policy for online estimation is replaced by expert demonstrations in a given dataset. For ICRL, The goal is to recover the minimum constraint set that best explains the expert data. Existing ICRL works commonly follow the Maximum Entropy framework (Malik et al., 2021). IRC solvers in this setting follow the same framework but solve a bi-level optimization problem (Hugessen et al., 2024).

We build on the codebases from Hugessen et al. (2024) and Liu et al. (2023) to compare the transferability performance of the IRC and ICRL solvers. We adapt the code to enable both solvers to infer constraint knowledge—such as reward correction terms or cost functions—within the source environment while evaluating the feasibility of this knowledge in the target environment. Using the blocked half-cheetah environment as a testbed, we report the results with mean $\pm$ standard deviation in Figure 7 with three random seeds. The definition of metrics and detailed source and target environment specifications are explained in Section E.

We find that IRC solver has better training efficiency in the source environment, i.e., achieving zeo violation rate with considerable feasible rewards more quickly than the ICRL solver. However, after transferring constraint knowledge into the target environment, inferred correction terms by the IRC solver fail to ensure safety (avoid constraint violation) while the cost function inferred by the ICRL solver has better generalizability.

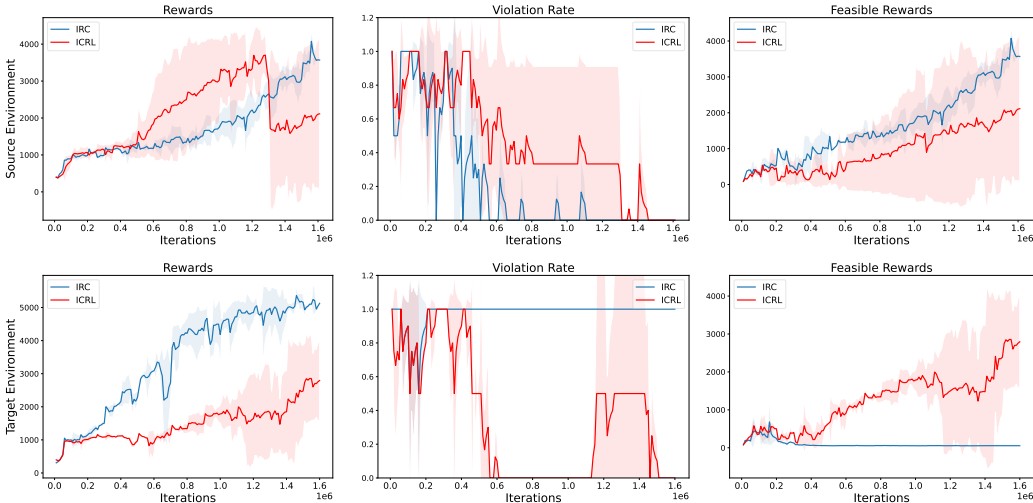

Figure 7: Training curves of rewards (left), violation rate (middle), and feasible rewards (right) for the ICRL (red) and IRC (blue) solvers. The top row shows the results for the source environment, and the bottom row shows the results for the target environment.

# E    EXPERIMENTAL DETAILS

We ran experiments on a desktop computer with Intel(R) Core(TM) i5-14400F and NVIDIA GeForce RTX 4060 Ti.

**Details about Gridworld.** In this paper, we construct a map with dimensions of $7 \times 7$ units and define four distinct settings, as shown in Figure 2. Locations are represented by two coordinates, with the first corresponding to the vertical axis and the second to the horizontal axis. The agent's objective is to navigate from a starting point to a target location while avoiding specified constraints. The agent begins in the bottom-left corner at position $(0, 0)$ and has eight possible actions: four cardinal directions (up, down, left, right) and four diagonal directions (upper-left, lower-left, upper-right, lower-right). The target location and reward are positioned in the upper-right cell $(6, 6)$ in the first, second, and fourth Gridworld environments, while in the third environment, the target is located in the upper-left cell $(6, 0)$. If the environment has a stochasticity of $p$, the agent has a probability of $p$ to move randomly in any feasible direction, with each direction having a probability of $p/\text{num\_of\_actions}$. The reward is only provided at the target cell, with all other cells yielding zero reward. A cost of $1$ is incurred if the agent enters a constrained location. Each policy rollout continues for a maximum of 50 time steps. In Figure 4, we present the mean and the 68% confidence interval (1-sigma error bar), calculated using three random seeds. Table 3 presents utilized hyperparameters in Gridworld experiments.

Table 3: List of the utilized hyperparameters in the Gridworld environment.

| Parameters | Gridworld |
| --- | --- |
| Max Episode Length | 50 |
| Discount Factor | 0.7 |
| Stopping Threshold | 0.001 |
| Stochasticity | 0.05 |
| Nu Max Clamp | 1 |
| Penalty Initial Value | 0.1 |
| Penalty Learning Rate | 0.1 |
| Source Terminal Rewards | 1,1,1,1 |
| Target Terminal Rewards | 2,7,7,15 |
| Ground Truth Costs | 1 |

**Details about Half-Cheetah** The Blocked Half-Cheetah task is built on Mujoco, where the agent controls a two-legged robot. The reward is determined by the distance the robot travels between consecutive time steps, penalized by the magnitude of the input action. Each episode ends after a maximum of 5000 time steps. To impose a constraint, we block the region where the X-coordinate should satisfy $x\_pos \le -3$, restricting the robot's movement to the region where the X-coordinate is between $-3$ and $+\infty$. The source environment follows the setup described above. In the target environment, rewards are scaled by a factor of $1.1$, and Gaussian noise with a mean of $0$ and a standard deviation of $0.1$ is added to each observation. We utilize three metrics for evaluations: 1) rewards are defined as the total returns for an episode, regardless of constraints; 2) feasible rewards are the aggregated returns for an episode up to the first constraint violation; 3) the violation rate is calculated as the percentage of episodes in which one or more constraint violations occur. Table 4 presents utilized hyperparameters in Gridworld experiments. Other hyperparameters follow previous codebases (Hugessen et al., 2024; Liu et al., 2023).

Table 4: List of the utilized hyperparameters in the Half-Cheetah environment.

| Parameters | Half-Cheetah |
| --- | --- |
| Training Epoch | 320 |
| Testing Epoch | 320 |
| Max Episode Length | 5000 |
| IRC Solver | IRL-base |

