# OpenReview forum: "Understanding Constraint Inference in Safety-Critical Inverse Reinforcement Learning"
_ICLR.cc/2025/Conference — ICLR 2025 Poster_

### Official Review · Reviewer_87Lh · 2024-11-02

**Soundness:** 3
**Presentation:** 3
**Contribution:** 3
**Rating:** 6
**Confidence:** 2

**Summary:**

This paper focuses on two approaches for safety-critical inverse RL: Inverse Constrained Reinforcement Learning (ICRL) and Inverse Reward Correction (IRC). It derives the upper bounds of these two methods and concludes the advantage of IRC in terms of sample efficiency. The paper also discusses potential constraint violations when learning a reward correction term or cost in source environments and transferring them to new environments. Theoretical analysis demonstrates that ICRL is more robust to these issues compared to IRC and examines the optimality of ICRL in target environments. Finally, empirical studies on gridworlds validate the theoretical findings for both methods.

**Strengths:**

- The paper provides a comprehensive comparison of ICRL and IRC approaches from multiple perspectives, including sample complexity and transferability. This reveals a tradeoff between the methods, offering interesting insights into their applications.
- The theoretical formulations and derivations are detailed and well-defined.
- The empirical evaluations are convincing and consistent with theoretical results.

**Weaknesses:**

See Questions.

**Questions:**

- The paper compares sample efficiency using theoretical upper bounds. I haven’t fully examined the details, but I am curious about the tightness of these estimations and whether they reliably represent real sample efficiency in practice.
- Although the experiments are conducted in gridworlds to validate the theoretical findings, I wonder if the results hold true for more complex, scalable tasks. Discussing potential challenges in such contexts could offer guidance and inspiration for the practical application of ICRL and IRC.
- I am a little confused about $\min_{\triangle r}$ in Eq. (2), which seems to minimize the expert's returns. Should it be $\max$ instead?
- The term 'iteration' appears in several theorems. It would be helpful to briefly explain what ICRL and IRC do in each iteration, such as taking a fixed number of gradient steps based on Eq. (1) and (2) ?

---

> ### Author Response · Authors · 2024-11-22
> **Author response to Reviewer 87Lh**
>
> Dear Reviewer 87Lh,
>
> Thank you for your thoughtful and comprehensive review of our manuscript. We have meticulously updated the manuscript, ensuring all revisions are clearly highlighted in orange for your convenience. We have given careful consideration to your suggestions and hope the following responses satisfactorily address your concerns.
>
> >Comment 1: The paper compares sample efficiency using theoretical upper bounds. I haven’t fully examined the details, but I am curious about the tightness of these estimations and whether they reliably represent real sample efficiency in practice.
>
> **Response 1:** Thanks for the suggestion. The upper bound is proposed to guide the exploration, and by exploring to minimize the upper bound of error, the solvers are theoretically guaranteed to have a better estimate of reward correction terms or costs. Besides, we consider an infinite horizon scenario where a lower bound is hard to derive.
>
> ---
>
> >Comment 2: Although the experiments are conducted in gridworlds to validate the theoretical findings, I wonder if the results hold true for more complex, scalable tasks. Discussing potential challenges in such contexts could offer guidance and inspiration for the practical application of ICRL and IRC.
>
> **Response 2:** Thanks for this comment. To address the extension to high-dimensional continuous environments, we incorporate the Maximum Entropy framework of ICRL solvers [1] and IRC solvers [2]. Additionally, our original online approach can be adapted to an offline setting, where the agent learns to infer constraint knowledge that best aligns with expert demonstrations from an offline dataset, rather than relying on a predefined expert policy for querying. Our results demonstrate that while ICRL exhibits slower learning in the training environment, it achieves superior transferability when environment rewards or transition dynamics differ in the target environment. This observation is consistent with our theoretical findings for discrete spaces. In the revised manuscript, we have detailed this addition in lines 483–486 (highlighted in orange). Further experimental results for comparison are provided in Appendices C and D.
>
> There are certain challenges for more complex environments. First, the offline dataset may contain noise or sub-optimal demonstrations, robustly learning constraint knowledge or identifying these annoying demonstrations holds promise. Second, because of the multi-level optimization layers, both solvers may take a long time to converge.
>
> We acknowledge that exploring real-world applications could provide deeper insights into the practical challenges and opportunities of transferring constraint information, ultimately guiding the development of more robust and scalable algorithms.
>
> ---
>
> > Comment 3: I am a little confused about $\min_{\mathit{\Delta r}}$
>  in Eq. (2), which seems to minimize the expert's returns. Should it be $\max$
>  instead?
>
> **Response 3:** Thanks for your comment. This is derived from Eq. (1) where $\mathit{\Delta r}=-\lambda c, \lambda>0$. Maximizing the objective with $c$ in Eq. (1) is equivalent to minimizing the objective with $\mathit{\Delta r}$, since the objective is linear with regard to $\mathit{\Delta r}$ or $c$.
>
> ---
>
> > Comment 4: The term 'iteration' appears in several theorems. It would be helpful to briefly explain what ICRL and IRC do in each iteration, such as taking a fixed number of gradient steps based on Eq. (1) and (2)?
>
> **Response 4:** Thanks for your comment. In each iteration, we first uniformly collect samples from the state-action space, and then update the estimation of the transition model and expert policy based on these samples. As the number of samples increases, the estimation error of transition and expert policy reduces. This leads to better estimation of either feasible reward correction terms or feasible cost functions. What the two solvers do in each iteration is detailed in Appendix Algorithm 1.
>
> ---
>
> References
>
> [1] Shehryar Malik, Usman Anwar, Alireza Aghasi, and Ali Ahmed. Inverse constrained reinforcement learning. In International Conference on Machine Learning (ICML), pp. 7390–7399, 2021.
>
> [2] Hugessen, Adriana, Harley Wiltzer, and Glen Berseth. "Simplifying constraint inference with inverse reinforcement learning." The Thirty-eighth Annual Conference on Neural Information Processing Systems. 2024.

---

> > ### Comment · Reviewer_87Lh · 2024-11-26
> >
> > Thank you for your response. I would like to confirm several points: For the first question, it is clear that the IRC provides a better upper bound. I am not familiar with the sample complexity theory of IRC and ICRL but is it fair to use the upper bounds to compare sample complexity and claim that the IRC outperforms IRCL in terms of sample complexity, especially when the tightness of this bound is unclear?
> >
> > For the third question, I would like to confirm whether the change of variable from $c$ to $\mathit{\Delta r}=-\lambda c$ would switch the optimization operation from $\max$ to $\min$. Because the objective inside seems to remain the same in Eq. (1) and (2).

---

> ### Author Response · Authors · 2024-11-27
> **Further Response on the 1st and 3rd Questions.**
>
> Dear Reviewer 87Lh,
>
> Thank you for your thoughtful feedback and for engaging in the discussion with us.
>
> > Comment: tightness of sample complexity upper bound for IRC and ICRL solvers
>
> **Response:** We agree with the reviewer that upper bounds are an important part of the analysis, and demonstrating the tightness of these bounds is critical for a more rigorous comparison. However, our work primarily focuses on the transferability of constraint knowledge, with an emphasis on discussing the safety and optimality of policies based on constraint knowledge in target environments. The main takeaway we want from previous related works of IRC and ICRL is that IRC offers a better upper bound in terms of sample complexity, which suggests it outperforms ICRL in this regard.
>
> In our analysis, based on Lemma 4.1 (implicit definition of the feasible reward correction set) and Lemma 4.3 (implicit definition of the feasible cost set), we show that ICRL solvers need to estimate further the advantage function for expert alignment, a step that is not needed for IRC solvers. This extra estimation step in ICRL leads to a higher sample complexity compared to IRC under uniform sampling from a generative model. Thus, we argue that the sample complexity of ICRL is, in theory, larger than that of IRC.
>
> Empirical experiments further support this theoretical finding. We observe that IRC consistently requires fewer samples than ICRL to achieve comparable performance levels in the source (learning) environments. This empirical result also matches the findings presented in [1].
>
> In our work, we adopt the feasible set approach for comparison of sample complexity upper bounds. This approach aims to recover the entire correction terms (for IRC) and cost functions (for ICRL) compatible with the demonstrations, thereby addressing the identifiability problem in IRL (or IRC/ICRL) without being influenced by the choice of heuristics or additional restrictions, as discussed in [2].
>
> To the best of our knowledge, the feasible set approach is a relatively new framework for analyzing sample complexity and this line of works only include [3], [4], and [5]. Specifically, [3] introduced this approach in the infinite horizon case for IRL, [4] derived a lower bound in the finite horizon setting with additional assumptions for IRL to assess the tightness of upper bounds, and [5] extended [3] to ICRL.
>
> We agree that proving the tightness of the upper bounds through lower bound analysis or other methods would be a valuable direction for future research, and we plan to explore this in subsequent work.
>
> References
>
> [1] Hugessen, A., et al. Simplifying constraint inference with inverse reinforcement learning. NeurIPS, 2024.
>
> [2] Lazzati, Filippo, Mirco Mutti, and Alberto Maria Metelli. How does Inverse RL Scale to Large State Spaces? A Provably Efficient Approach. NeurIPS, 2024.
>
> [3] Alberto Maria Metelli, Giorgia Ramponi, Alessandro Concetti, and Marcello Restelli. Provably efficient learning of transferable rewards. ICML, 2021.
>
> [4] Alberto Maria Metelli, Filippo Lazzati, and Marcello Restelli. Towards theoretical understanding of inverse reinforcement learning. ICML, 2023.
>
> [5] Yue, Bo, Jian Li, and Guiliang Liu. Provably Efficient Exploration in Inverse Constrained Reinforcement Learning. arXiv preprint arXiv:2409.15963.
>
> ---
>
> > Comment: the third question
>
> **Response:** Thank you for reminding us; you are definitely correct. Since the objective function inside remains the same and the variable $\mathit{\Delta r}$ falls into the domain $-\lambda c$, it should be $\max_{\mathit{\Delta r}}$ instead of $\min_{\mathit{\Delta r}}$. Our previous response was inaccurate, and we sincerely apologize for the oversight. We have corrected this typo in the revised manuscript. This does not affect later analyses of IRC and ICRL solvers.
>
> Thank you again for your valuable comments. We would be more than happy to continue the discussion if there are any further questions.

---

> > ### Comment · Reviewer_87Lh · 2024-11-28
> >
> > Thank you for your response. The clarification of the theoretical motivation and the related work in your response helps me better understand your work. I choose to maintain my score.

---

> > > ### Author Response · Authors · 2024-11-29
> > > **Author Response to Reviewer 87Lh**
> > >
> > > Dear Reviewer 87Lh,
> > >
> > > We are deeply grateful for the reviewer’s feedback and the significant time and effort invested in reviewing our manuscript. Your insightful comments have been invaluable in enhancing the clarity and quality of our work.
> > >
> > > Thank you very much! Wishing you a joyful Thanksgiving!

---

### Official Review · Reviewer_i1Qn · 2024-11-04

**Soundness:** 3
**Presentation:** 4
**Contribution:** 3
**Rating:** 6
**Confidence:** 2

**Summary:**

The paper explores inverse reward correction (IRC) that, instead of solving a three-level optimization in inverse constrained RL (ICRL), tries to combine the constraints as part of the reward function so as to solve only a reward optimization problem. The paper proves IRC has lower sample complexity but worse generalizability than ICRL solvers. The work investigates conditions in the transition laws, cost/reward functions, and Lagrangian multipliers that guarantee epsilon-optimality of IRC when transferring to a different environment.

**Strengths:**

- well-written and clear
- problem and questions are well-motivated
- interesting theoretical guarantees and analysis of IRC

**Weaknesses:**

- could benefit from comparison on more complex environments than grid-world
- it would be nice to have a comparison of the pseudocode of the IRC and ICRL algorithms used in experiment
- consider adding hyperparameters of the algorithms in the experiments

**Questions:**

- what are the implications of this in few-shot learning after transferring an IRC/ICRL policy?
- What would these bounds on the sample complexity/generalizability of IRC and ICRL mean in real world tasks where S and A are continuous spaces?
- Furthermore, do these bounds consider any measure (like complexity) of the dynamics/transition function? Why or why not?

---

> ### Author Response · Authors · 2024-11-22
> **Author response to Reviewer i1Qn - Part 1**
>
> Dear Reviewer i1Qn,
>
> We deeply value your detailed review of our manuscript and the constructive feedback you provided. To address your comments, we have revised the manuscript, with all modifications distinctly highlighted in orange for ease of review. We have thoroughly considered your suggestions and wish that the responses below will resolve your concerns.
>
> > Comment 1: Could benefit from comparison on more complex environments than grid-world
>
> **Response 1:** Thank you for raising this concern. To address the extension to high-dimensional continuous environments, we incorporate the Maximum Entropy framework of ICRL solvers [1] and IRC solvers [2]. Additionally, our original online approach should be adapted to an offline setting, where the agent learns to infer constraint knowledge that best aligns with expert demonstrations from an offline dataset, rather than relying on a predefined expert policy for querying. Our results demonstrate that while ICRL exhibits slower learning in the training environment, it achieves better transferability when environment rewards or transition dynamics differ in the target environment. This observation is consistent with our theoretical findings for discrete spaces. In the revised manuscript, we have detailed this addition in lines 483–486 (highlighted in orange). Further experimental results for comparison are provided in Appendices C and D.
>
> In addition to conducting empirical studies, we have made considerable contributions to understanding the differences among various solvers used for modeling constraint knowledge in the field of constraint inference.
>
> ---
>
> > Comment 2: It would be nice to have a comparison of the pseudocode of the IRC and ICRL algorithms used in the experiment.
>
> **Response 2:** The pseudocode is provided in Algorithm 1 in Appendix Section B.3. We omitted it from the main body of the original draft due to the page limit. A comparison of pseudo-code is now available in lines 761-764, highlighted in orange. We have explained $\mathcal{I}^{\mathit{\Delta r}}_ {k+1}$ in lines 226-234 and $\mathcal{I}^{c}_ {k+1}$ in lines 264-269 in the revised manuscript.
>
> ---
>
> > Comment 3: Consider adding hyperparameters of the algorithms in the experiments.
>
> **Response 3:** Thanks for this comment. In the original draft, we illustrate the utilized hyperparameters in lines 1332-1346 in Appendix Section C. To enhance rigor and clarity, we have added a list of utilized hyperparameters in Appendix Table 3 and 4 in the revised version (table caption highlighted orange).
>
> ---
>
> > Comment 4: What are the implications of this in few-shot learning after transferring an IRC/ICRL policy?
>
> **Response 4:** Instead of transferring policies, we focus on transferring feasible constraint knowledge, such as reward correction terms or cost functions, from the source to the target environment. The agent derives this knowledge entirely from the source and uses it to generate a policy in the target environment, aligning with a zero-shot learning paradigm. This approach emphasizes the generalizability of IRC and ICRL solvers, distinguishing them from imitation learning methods. Implications are that although IRC has lower training efficiency to ensure the optimality of expert agents in source environments, it has poorer performance at generalizability when transferring.

---

> ### Author Response · Authors · 2024-11-22
> **Author response to Reviewer i1Qn - Part 2**
>
> >Comment 5: What would these bounds on the sample complexity/generalizability of IRC and ICRL mean in real world tasks where S and A are continuous spaces?
>
> **Response 5:** Thanks for this comment. Sample complexity analysis has primarily focused on discrete state-action spaces [3].
> Extending such analyses to continuous spaces remains a significant challenge in the field. Existing algorithms for learning feasible sets [4, 5, 6] face difficulties when scaling to problems with large or continuous state spaces. This is largely due to their sample complexity being directly tied to the size of the state space, which presents a substantial limitation since real-world problems often involve large or continuous spaces.
> Continuous domains often require function approximation techniques, additional assumptions (e.g. for smoothness or linear structures), and more sophisticated exploration strategies. Generalizability is another concern due to the infinite nature of these domains since it relies heavily on the ability to approximate value functions, policies, and constraints. We leave developing a scalable approach for sample complexity analysis as our future work.
>
> ---
>
> > Comment 6: Furthermore, do these bounds consider any measure (like complexity) of the dynamics/transition function? Why or why not?
>
> **Response 6:** Yes, they do. The complexity of the transition function is determined by the cardinality of the state-action space since the transition is a mapping from state-action to state: $\mathcal{S}\times \mathcal{A}\rightarrow \mathcal{S}$. The size of the transition matrix is $O(\mathcal{S}^2\mathcal{A})$. This bound relies on the size of state space $\mathcal{S}$ and the size of action space $\mathcal{A}$.
>
> ---
>
> References
>
> [1] Shehryar Malik, Usman Anwar, Alireza Aghasi, and Ali Ahmed. Inverse constrained reinforcement
> learning. In International Conference on Machine Learning (ICML), pp. 7390–7399, 2021.
>
> [2] Hugessen, Adriana, Harley Wiltzer, and Glen Berseth. "Simplifying constraint inference with inverse reinforcement learning." The Thirty-eighth Annual Conference on Neural Information Processing Systems. 2024.
>
> [3] Agarwal, Alekh, et al. "Reinforcement learning: Theory and algorithms." CS Dept., UW Seattle, Seattle, WA, USA, Tech. Rep 32 (2019): 96.
>
> [4] Alberto Maria Metelli, Filippo Lazzati, and Marcello Restelli. Towards theoretical understanding of inverse reinforcement learning. ICML, 2023.
>
> [5] Lei Zhao, Mengdi Wang, and Yu Bai. Is inverse reinforcement learning harder than standard reinforcement learning? ICML, 2024.
>
> [6] Filippo Lazzati, Mirco Mutti, and Alberto Maria Metelli. Offline inverse rl: New solution concepts and provably efficient algorithms. ICML, 2024.

---

> > ### Comment · Reviewer_i1Qn · 2024-11-26
> >
> > Thank you for your response. I maintain my score.

---

> > > ### Author Response · Authors · 2024-11-29
> > > **Author response to Reviewer i1Qn**
> > >
> > > Dear Reviewer i1Qn,
> > >
> > > We are delighted by your positive assessment of our work and are most grateful for your thoughtful and thorough review of our manuscript. Your insightful comments have been invaluable in improving the clarity and precision of our work. We appreciate the significant time and effort you have dedicated to this process. Thank you very much! Wishing you a happy Thanksgiving!

---

### Official Review · Reviewer_foPH · 2024-11-05

**Soundness:** 2
**Presentation:** 1
**Contribution:** 3
**Rating:** 5
**Confidence:** 5

**Summary:**

The paper addresses the task of inferring safety constraints from expert demonstrations where the expert maximizes a known reward subject to unknown constraints. Although inverse constrained reinforcement learning (ICRL) has developed specialized methods for this task, recent work has shown that casting the problem into the framework of inverse reinforcement learning (IRL) via "inverse reward correction" (IRC) could offer advantages. This paper provides a deeper analysis of these claims, showing that:
1) Inverse reward correction can achieve better sample efficiency than ICRL for constraint inference;
2) The implicit constraints inferred through IRC, however, may transfer poorly across environments with different dynamics or reward structures compared to constraints explicitly modeled by ICRL; and
3) ICRL’s explicit constraint modeling offers conditions under which its inferred constraints allow for approximately optimal and safe policies when transferred to new environments, even under varied dynamics.

**Strengths:**

- the task of inferring unknown safety constraints is important
- the recent challenge that the field of ICLR with its growing literature may be redundant deserves further scrutiny, and the results in this paper add important nuance to this challenge, showing that such redundancy is not clear cut
- the paper shows potentially useful results relating inverse reward correction and ICRL that deserve a place in the literature

**Weaknesses:**

- unfortunately, the presentation is bad, making deciphering the paper a painful challenge even to a motivated reader closely familiar with the work this paper builds on:
    - for me, this point alone currently warrants rejection, but I may be swayed if the authors manage to upload a substantial revision
    - here is a non-exhaustive list of issues:
    	- for $Q^{r,\pi}_{\mathcal{M}\cup r'}$ (and similarly V), it's never made clear how the value depends on $r$ in the superscript, which remains confusing throughout, especially as other values including the cost c get swapped into that place and this value is quite central in the paper and used frequently throughout the text
    	- the paper initially introduces a soft-constraint setting (allowing for costs up to $\epsilon\geq 0$, with the hard-constraint case $\epsilon=0$ as a special case) but then, in places, assumes the hard-constraint setting without alerting the reader to the fact adding further confusion
    	- several pieces of notation are not defined in the main text (e.g. $N^+_{k+1}$) on page 5
    	- l.219 mentions advantage, followed by a definition of what appears to be a state-value function instead with advantage never defined

	- regarding minor issues:
		- I'd advise using similar notation for different concepts: e.g. depending on sub/superscript, $\mathcal{C}$ is sometimes a set of feasible cost functions, sometimes a (not directly related) scalar value in the sample complexity bound, or $\Delta$ is once a set of measures, another time a reward correction term. I'd advise using at least a different font in the two cases.
		- several times, the main text refers to numbered Tables or Figures, which are in the appendix but this is not pointed out - by default, a reader will search in the main text
		- there is also a fair amount of typos

- the theoretical results very closely follow similar prior results on the sample complexity of IRL and the transferrability of the recovered rewards (Metelli 2021, Schlaginhaufen 2024). Per se, I don't see that as a huge problem - clearly porting these results to the context of constraint inference still has value. However, if the main contribution of the paper is translating the results into another context, presentation would seem to be one of the main possible contributions, so doing a poor job at that takes away most of the value

**Questions:**

- what role does the superscript $r$ play in  $Q^{r,\pi}_{\mathcal{M}\cup r'}$  ?
- in Definition 3.3 of an IRC solver, you introduce $\mathcal{C}_{\text{IRL}}$, "the set of feasible reward correction terms derived by [the solver]", which seems to imply that the IRC solver at least *may* recover a whole set of feasible reward correction terms. Then, having this set, one may try to produce a policy that is as robust as possible with respect to that set. Using your own example from Fig. 3, you write that the solver learns a error correction term $-1-\beta,\beta>0$, If understood as a set of feasible correction terms, a robust policy would avoid this state. Instead, you seem to force a choice of one particular reward correction. What do you think of using the full set of feasible error correction terms? Would this eliminate the advantage you point out? Alternatively, Bayesian IRL methods would give a posterior over the different feasible error correction terms, again allowing for producing robust policies.
- could you please point out which results and proofs mostly just follow prior work on related concepts, and which parts (e.g. giving line ranges) are indeed novel and specific to this context?

---

> ### Author Response · Authors · 2024-11-22
> **Author response to Reviewer foPH - Part 1**
>
> Dear Reviewer foPH,
>
> We sincerely appreciate your thorough, valuable, and constructive review of our manuscript. We have carefully revised the manuscript, with all changes highlighted in orange for clarity. Your suggestions have been thoughtfully considered, and we hope the following responses adequately address your concerns.
>
> > Comment 1: For $Q^{r,\pi}_{\mathcal{M}\cup r^\prime}$ (and similarly V), it's never made clear how the value depends on $r$ in the superscript, which remains confusing throughout, especially as other values including the cost c get swapped into that place and this value is quite central in the paper and used frequently throughout the text.
>
> **Response 1:** We apologize for any potential ambiguity in the original manuscript. This notation is actually clarified in lines 218–222 under Lemma 4.1 in the original draft, where it is first introduced. The *superscript* $r$ (or $c$) in $Q^{r,\pi^E}_ {\mathcal{M} \cup r_1}$ (or $Q^{c,\pi^{E}}_{\mathcal{M} \cup c_1}$) specifies whether the function represents a reward $Q$-function or a cost $Q$-function, highlighting the distinct roles of the reward or cost value functions in constraint inference. The *subscript* $r_1$ (or $c_1$) identifies the specific rewards or costs being evaluated by the $Q$-function.
>
> For greater clarity, we have now explicitly defined this notation in lines 150–156 of the revised manuscript.
>
> ---
>
> > Comment 2: The paper initially introduces a soft-constraint setting (allowing for costs up to $\epsilon\geq 0$, with the hard-constraint case $\epsilon=0$ as a special case) but then, in places, assumes the hard-constraint setting without alerting the reader to the fact adding further confusion.
>
> **Response 2:** We understand that the reviewer is likely referring to Section 5. In the original manuscript, we specified where a hard-constraint setting is assumed: in line 372 for Section 5.1 (safety) and line 384 for Section 5.2 (optimality).
>
> To improve clarity and rigor, as recommended by the reviewer, we have explicitly identified the hard-constraint setting in line 346, as well as in Lemma 5.2 and Theorem 5.3, in the revised manuscript.
>
> Also, please note that we have studied the soft constraint scenario in Appendix B.8.2, as mentioned in the paragraph 'Extension to Soft Constraint'.
>
> ---
>
> > Comment 3: Several pieces of notation are not defined in the main text (e.g. $N^+_{k+1}$) on page 5.
>
> **Response 3:** Thanks for this comment. These definitions were initially provided in the appendix due to space limitations in the main text. To improve clarity, we have now included explanations for the significance $\delta$ and the cumulative count of visitations to $N^+_{k+1}$ in lines 227 and 236 of the revised manuscript, respectively.
>
> ---
>
> > Comment 4: l.219 mentions advantage, followed by a definition of what appears to be a state-value function instead with advantage never defined.
>
> **Response 4:** Thanks for your correction. For better clarity, we have now clearly defined the reward advantage function in line 154 of the revised manuscript.
>
> ---
>
> > Comment 5: I'd advise using similar notation for different concepts: e.g. depending on sub/superscript, $\mathcal{C}$ is sometimes a set of feasible cost functions, sometimes a (not directly related) scalar value in the sample complexity bound, or $\Delta$ is once a set of measures, another time a reward correction term. I'd advise using at least a different font in the two cases.
>
> **Response 5:** We greatly appreciate this valuable feedback. In response, we have updated the scalar values in the sample complexity bound, by replacing $\mathcal{C}^c_{k+1}$ with $\mathcal{I}^c_{k+1}$ and replacing $\mathcal{C}^ {\mathit{\Delta r}}_ {k+1}$ with $\mathcal{I}^{\mathit{\Delta r}}_{k+1}$, respectively. Furthermore, we have modified the notation for the reward correction term, changing $\Delta r$ to $\mathit{\Delta r}$ for clarity and consistency. We have highlighted this modification in orange in the revised manuscript.
>
> ---
>
> > Comment 6: Several times, the main text refers to numbered Tables or Figures, which are in the appendix but this is not pointed out - by default, a reader will search in the main text
>
> **Response 6:** We apologize for the oversight of not directly referring to "Appendix" for Table 1 and Algorithm 1. This has now been corrected in the revised manuscript in line 140 and line 265. Regarding the reference to Figure 1 under Theorem 5.3, we believe this refers to Figure 1 in the main text. We have thoroughly reviewed the draft to eliminate this issue.
>
> ---
>
> > Comment 7: There is also a fair amount of typos.
>
> **Response 7:** Thanks for your correction. We have conducted a thorough review of the draft and prepared a revised version with the corrected parts highlighted in orange for clarity.

---

> > ### Comment · Reviewer_foPH · 2024-11-22
> >
> > Thank you for the responses! Here are a few more comments in reaction to those
> >
> > Response 1: Thank you for the clarification. However, since $r$ is also used as a variable name in the text (even in the same font!), the superscript suggests that the value of $Q^r$ depends on the value of the variable $r$ via the superscript. Two possible ways to fix this come to mind: replace $r$ with $\text{rew}$ in the superscript or something like that to clearly distinguish it from a variable $r$ (at the very least, one should use a different font (e.g. $Q^{\text{r}}$ but I think that could still remain confusing). Or just use a different letter for $Q^c$ altogether, such as $C$.
> >
> > However, this brings another question: if the superscript is used to distinguish this from $Q^c$, then is the definition meaningfully different (also note that $Q^c$ is never defined in the main text). I'd say it isn't, if you use the right notation. E.g. If you use e.g. $Q^{r,\pi}_{\mathcal{M}}$ to actually denote the expected return w.r.t. $r$ (or $c$ swapped into the same spot), then you can just use a single definition and get rid of the confusion.
> >
> > Response 2: Yes, I noticed that, but the unfortunate choice is that you first state one assumption (possibly soft constraints). Then you give a bunch of results (at odds with that assumption, which confuses the reader). And only then you reveal to them that you were in fact using another assumption. If you're changing an assumption, please state that before results building on that assumption.
> >
> > Response 3-7: Thank you for the responses and improving the manuscript.

---

> ### Author Response · Authors · 2024-11-22
> **Author response to Reviewer foPH - Part 2**
>
> > Comment 8: What role does the superscript $r$ play in $Q^{r,\pi}_{\mathcal{M}\cup r^\prime}$?
>
> **Response 8:** Please refer to Response 1.
>
> ---
>
> > Comment 9: In Definition 3.3 of an IRC solver, you introduce $\mathcal{C}_{\text{IRC}}$, "the set of feasible reward correction terms derived by [the solver]", which seems to imply that the IRC solver at least may recover a whole set of feasible reward correction terms. Then, having this set, one may try to produce a policy that is as robust as possible with respect to that set. Using your own example from Fig. 3, you write that the solver learns an error correction term $-1-\beta,\beta>0$. If understood as a set of feasible correction terms, a robust policy would avoid this state. Instead, you seem to force a choice of one particular reward correction. What do you think of using the full set of feasible error correction terms? Would this eliminate the advantage you point out? Alternatively, Bayesian IRL methods would give a posterior over the different feasible error correction terms, again allowing for producing robust policies.
>
> **Response 9:** Instead of transferring robust policies, we transfer reward correction terms or cost functions from source to target. This is a key distinction between inverse learning algorithms such as IRL, IRC, and ICRL, and imitation learning approaches like behavior cloning.
>
> In the source environment, we infer a feasible set of reward corrections or costs that align with the given expert policy. Policies in the target environment are then derived based on these inferred terms. For example, in Fig. 3, we demonstrate a subset of the feasible set ($-1-\beta, \beta>0$) that leads to unsafe policies in the target environment, although it leads to safe policies in the source environment. Moving one step further, we propose Theorem 5.3 to precisely identify the conditions under which such "unsafe" feasible terms can arise.
>
> ---
>
> > Comment 10: Could you please point out which results and proofs mostly just follow prior work on related concepts, and which parts (e.g. giving line ranges) are indeed novel and specific to this context?
>
> **Response 10:** Yes, we list them as follows. They have also been specified in the contribution part of the introduction. The part of ICRL solvers recaps previous work. We formalize and summarize the IRC solvers and follow the prior theoretical framework to analyze its sample complexity (lines 211-245). We make a comparison between the two solvers (IRC vs. ICRL) and discuss the source of additional sample complexity for ICRL (lines 277-286).
> We propose and investigate the safety issues in transferability (lines 300-377). Concerning the optimality issues, we extend the transferability definition from prior work in MDP settings to accommodate more complicated CMDP settings. We derive conditions that limit the similarity between source and target environments to ensure $\varepsilon$-optimality for ICRL (lines 420-452). We empirically validate our results on training efficiency and cross-environment transferability in various environments (lines 453-519).

---

> > ### Comment · Reviewer_foPH · 2024-11-22
> >
> > **Response 9**: Sorry, you may have misunderstood my point, so let me clarify. You have a whole set of feasible correction terms. Some of these may be "unsafe", though when talking about an unsafe correction term, we really mean an unsafe policy. So we have a set of feasible policies, some of which are unsafe. But if there are some that are robustly safe? Then, if we indeed care about prioritizing safety, we would choose a policy that is robustly safe, i.e. does not incur high cost under any of the recovered costs (/reward correction terms). If we choose such a policy, instead of an arbitrary one, you criticism of IRC as less safe than ICRL stops being sound, doesn't it?

---

> ### Author Response · Authors · 2024-11-24
> **Further Author Response for Discussion**
>
> Dear Reviewer foPH,
>
> Thank you for engaging in the discussion and giving us the opportunity to offer further clarification.
>
> > Additional Comment on 1: Confusion on notation $Q^{r,\pi}_ {\mathcal{M}\cup r^\prime}$ and $Q^{c,\pi}_ {\mathcal{M}\cup c}$.
>
> **Further response on on 1:** Thanks for this additional comment and valuable suggestion. We believe it requires the superscript to differentiate Q-functions of reward or cost under a CMDP $\mathcal{M}\cup c$.
> If we replace both $Q^{r,\pi}_ {\mathcal{M}\cup c}$ and $Q^{c,\pi}_ {\mathcal{M}\cup c}$ with $Q^{\pi}_ {\mathcal{M}\cup c}$, we will not be able to know whether $Q^{\pi}_ {\mathcal{M}\cup c}$ denotes the expected rewards or costs under $\mathcal{M}\cup c$.
> Based on your advice, we now figure out a clearer notation to address this confusion. There are a total of five notations under IRC and ICRL solvers: 1) for IRC, $Q^{r, \pi}_ {\mathcal{M}\cup (r+\mathit{\Delta r})} / Q^{\mathit{\Delta r}, \pi}_ {\mathcal{M}\cup (r+\mathit{\Delta r})} / Q^{r+\mathit{\Delta r}, \pi}_ {\mathcal{M}\cup (r+\mathit{\Delta r})}$, 2) for ICRL, $Q^{r, \pi}_ {\mathcal{M}\cup c} / Q^{c, \pi}_ {\mathcal{M}\cup c}$. The subscript specifies the environment $\mathcal{M}$ with either an updated reward function $r+\mathit{\Delta r}$ or a cost function $c$. The superscript specifies the actual rewards or costs under evaluation. If the superscript and the subscript are the same (e.g., in $Q^{c, \pi}_{\mathcal{M}\cup c}$), this means we are calculating the cumulative costs in the environment. We list them as follows.
>
> $Q^{r,\pi}_ {\mathcal{M}\cup (r+\mathit{\Delta r})}(s,a) = \mathbb{E}_ {\pi,P_\mathcal{T}}\left[\sum_{t=0}^{\infty} \gamma^t r(s_t, a_t)\right],$
>
> $Q^{\mathit{\Delta r},\pi}_ {\mathcal{M}\cup (r+\mathit{\Delta r})}(s,a) = \mathbb{E}_ {\pi,P_\mathcal{T}}\left[\sum_{t=0}^{\infty} \gamma^t \mathit{\Delta r}(s_t, a_t)\right],$
>
> $Q^{r+\mathit{\Delta r},\pi}_ {\mathcal{M}\cup (r+\mathit{\Delta r})}(s,a) = \mathbb{E}_ {\pi,P_\mathcal{T}}\left[\sum_{t=0}^{\infty} \gamma^t [r+\mathit{\Delta r}](s_t, a_t)\right],$
>
> $Q^{r,\pi}_ {\mathcal{M}\cup c}(s,a) = \mathbb{E}_ {\pi,P_\mathcal{T}}\left[\sum_{t=0}^{\infty} \gamma^t r(s_t, a_t)\right],$
>
> $Q^{c,\pi}_ {\mathcal{M}\cup c}(s,a) = \mathbb{E}_ {\pi,P_\mathcal{T}}\left[\sum_{t=0}^{\infty} \gamma^t c(s_t, a_t)\right]$.
>
> Note that although $Q^{r,\pi}_ {\mathcal{M}\cup (r+\mathit{\Delta r})}(s,a)$ and $Q^{r,\pi}_ {\mathcal{M}\cup c}(s,a)$ are equal in value, they belong to different solvers and the subscript specifies which solver each reward Q-function belongs to.
>
>
> We have revised the manuscript accordingly to incorporate this modification in the main text and appendix.
>
> ---
>
> > Additional Comment on 2: Mention the assumption to avoid confusion.
>
> **Further response on on 2:** We agree with the reviewer on this point. As mentioned in the previous rebuttal, in the revised manuscript, we have now explicitly added the assumption of hard-constraint settings in line 346, as well as in Lemma 5.2 and Theorem 5.3.
>
> ---
>
> > Additional Comment on 9: Comparison of IRC and ICRL on safety issues.
>
> **Further response on 9:**
> Thank you for your comment. We would like to provide additional clarification. By the word 'safe', we mean whether the inferred reward correction terms induce safe policies in the *target* environment. Labeling a correction term with 'safe' or 'unsafe' requires prior knowledge of the transition and reward functions in the target environment. However, in practice, we do not assume that such prior knowledge is available, i.e., we only know that the correction term aligns with expert policy in the source environment. This unavailability of prior knowledge is reasonable since the agent could directly infer the correction terms in the target environment if the agent had access to this additional knowledge. This makes any terms inferred from the source environment unnecessary. Theorem 5.3 states the conditions under which unsafe correction terms exist (safe in the source but unsafe in the target) and Figure 1 is one instance of such unsafe terms.
>
> In essence, each reward correction term is safe regarding a set of (reward, transition) pairs. Exploring solutions to identify the most robustly safe correction terms (encompassing the largest subset of a given set of reward-transition pairs) is an interesting direction for future research. However, this lies outside the scope of our current settings.
>
> If you have any further questions, we would be more than happy to engage in discussions.

---

> ### Author Response · Authors · 2024-11-29
> **Follow-up on Additional Response**
>
> Dear Reviewer foPH,
>
> We hope this message finds you well and that you're enjoying a wonderful Thanksgiving season!
>
> We sincerely appreciate the time and effort you have dedicated to reviewing our submission, especially during this busy period. We wanted to kindly follow up and invite you to review our additional response at your convenience. Your feedback is invaluable to us and will be instrumental in further refining our work.
>
> If there are any points that require clarification or if you would like further details, please don't hesitate to reach out. We are more than happy to provide any additional information you may need.
>
> Thank you again for your thoughtful insights and support. We truly appreciate your contributions to the review process.
>
> Wishing you a joyful and fulfilling Thanksgiving!
>
> Best regards,
>
> The Authors of Paper 6391

---

> ### Author Response · Authors · 2024-12-02
> **A kind reminder regarding our further response**
>
> Dear Reviewer foPH,
>
> We hope this message finds you well. We greatly appreciate the time and effort you have dedicated to reviewing our paper, especially given the many responsibilities you manage.
>
> As the discussion period is nearing its conclusion, we would like to kindly follow up regarding your feedback on our response. Your input is invaluable to us, and we are eager to address any further questions or concerns you may have.
>
> Thank you once again for your thoughtful contribution to this process. We deeply appreciate your time and expertise.
>
> Best regards,
>
> The Authors of Paper 6391

---

> > ### Comment · Reviewer_foPH · 2024-12-03
> >
> > Dear authors,
> > thank you for the substantial effort you put into clarifying your work and updating the manuscript. As a result, I think it's fair to increase my score, and I don't have any additional clarifying questions at this point. That said, I think the manuscript still remains slightly below the threshold I'd expect from papers at ICLR mostly due to the significance of the contributions and the overall clarity of presentation. In preparing possible future versions or other future work, I would recommend working with an editor or a colleague otherwise not involved in the project to polish the clarity of the paper prior to initial submission - I found my review work was initially made much harder than it could have been and I would have been able to provide better feedback on the substance on the paper had I started with an easier-to-read manuscript. That said, I'm a fan of this research direction and wish you all best in pursuing it further, if you choose so!

---

> > > ### Author Response · Authors · 2024-12-03
> > >
> > > Dear Reviewer foPH,
> > >
> > > We would like to express our sincere gratitude for your continued engagement, detailed feedback, and insightful discussions throughout this review process. We greatly appreciate the time, effort, and expertise you have dedicated to evaluating our paper.
> > > We will continue refining the paper based on your valuable comments.
> > >
> > > We also want to thank you for considering to increase the score. However, it appears that the revised score was not updated in the initial review.
> > >
> > > Thank you once again for your thoughtful contributions.
> > >
> > > Best regards,
> > >
> > > The Authors of Paper 6391

---

### Official Review · Reviewer_MoLg · 2024-11-10

**Soundness:** 3
**Presentation:** 3
**Contribution:** 3
**Rating:** 6
**Confidence:** 3

**Summary:**

This work introduces the IRC solver to overcome the limitation of the IRL solver, which generally lacks a mechanism to leverage existing reward signals and may not be compatible with different rewards. The authors also give theoretical analysis and achieve a lower sample complexity. Besides, they also study and extend the transferability. The results of the experiments demonstrate its efficiency.

**Strengths:**

1. This paper is well-written and easy to follow.
2. The related work is well organized.
3. The authors give a strong theoretical. analysis of the advantages and shortcomings between IRC and ICRL solvers, i.e., convergence, and sample complexity. Besides, the authors clearly presented their theory contributions.

**Weaknesses:**

1. What are the wall-clock running times of your method and the other baselines in the experiments?
2. In Theorem 3.2, Can we have such that directly maximizing the cumulative rewards and considering the constrained optimization objective?
3. In B.8.2 subsection, Can we provide experimental results to evaluate the soft constraint scenario?

**Questions:**

Please see the weaknesses.

---

> ### Author Response · Authors · 2024-11-22
> **Author response to Reviewer MoLg - Part 1**
>
> Dear Reviewer MoLg,
>
> We sincerely thank you for your detailed and insightful review of our manuscript. In response, we have carefully revised the manuscript, highlighting all changes in orange for discrepancies. We have thoughtfully incorporated your suggestions and believe that the following responses address your concerns effectively.
>
> > Comment 1: What are the wall-clock running times of your method and the other baselines in the experiments?
>
> **Response 1:** The table below reports the average wall-clock running times (mean $\pm$ standard deviation) in the format of minutes:seconds, based on 5 parallel experiments. The label 'source' indicates that the solver (ICRL or IRC) infers the reward correction term or cost function and applies it within the same environment as the source environment. In contrast, the label 'target' represents inference in the source environment following knowledge transfer to a distinct target environment. All experiments were conducted on a desktop computer equipped with an Intel(R) Core(TM) i5-14400F processor and an NVIDIA GeForce RTX 4060 GPU, consistent with the device specifications detailed in Appendix C.
>
> | Solver\Gridworld     | Gridworld-1       | Gridworld-2       | Gridworld-3       | Gridworld-4       |
> |----------------------|-------------------|-------------------|-------------------|-------------------|
> | ICRL-source          | 11:24 ± 00:06    | 09:12 ± 00:08    | 09:54 ± 00:09    | 09:39 ± 00:07    |
> | ICRL-target          | 12:36 ± 00:10    | 08:54 ± 00:08    | 11:18 ± 00:11    | 12:42 ± 00:12    |
> | IRC-source           | 08:27 ± 00:05    | 08:06 ± 00:05    | 08:03 ± 00:04    | 08:03 ± 00:04    |
> | IRC-target           | 08:12 ± 00:06    | 07:57 ± 00:05    | 08:15 ± 00:07    | 08:09 ± 00:06    |
>
> ---
>
> > Comment 2: In Theorem 3.2, Can we have such that directly maximizing the cumulative rewards and considering the constrained optimization objective?
>
> **Response 2:** Yes, there are algorithms that address the Constrained Reinforcement Learning (CRL) problem in a more direct manner. Below, we summarize these approaches:
>
> 1. Manual Selection of Lagrange Multipliers:
>    Methods such as [1, 2, 3] manually select Lagrange multipliers to directly maximize the objective $r - \lambda c$, where $r$ represents the reward and $c$ represents the cost.
>
> 2. Projection-Based Constraint Enforcement:
>    The approach in [4] utilizes prior knowledge of system transitions to project the policy's chosen action onto a set that guarantees constraint satisfaction, ensuring compliance without compromising the task.
>
> 3. Projection-Based Constrained Policy Optimization (PCPO):
>    PCPO [5] is an iterative algorithm designed to optimize policies under expected cumulative constraints. It operates in two stages:
>    - Stage 1: Maximizes the reward using TRPO [6], producing an intermediate policy that may not satisfy the constraints.
>    - Stage 2: Projects this intermediate policy onto the closest feasible policy, ensuring constraint satisfaction while improving the reward.
>    This method effectively balances reward optimization and constraint enforcement.
>
> While these methods offer innovative solutions, they come with limitations:
> - The first two approaches require additional information, such as system manual selected Lagrange multipliers or transition models.
> - PCPO has the drawbacks of being computationally expensive and has limited generality [7].
>
> In contrast, the Lagrangian relaxation method, the most widely adopted approach for addressing cumulative constraints [7], demonstrates high performance. This method achieves high long-term rewards and maintains low cumulative costs [8, 9].
>
> Theoretical support for this approach is provided by Theorem 3.2 from Paternain et al. (2019), which states that if the reward and cost functions are bounded, the constrained optimization problem (PI) can be solved precisely in the dual domain. This implies that the optimal policy for the constrained objective in (PI) can be obtained precisely by solving its dual problem (DI), which involves iteratively optimizing both the Lagrange multipliers and the policy.

---

> ### Author Response · Authors · 2024-11-22
> **Author response to Reviewer MoLg - Part 2**
>
> > Comment 3: In B.8.2 subsection, Can we provide experimental results to evaluate the soft constraint scenario?
>
> **Response 3:**  Yes, we have added additional results and analyses for the soft constraint scenario, as detailed at the end of Appendix Section B.8.2 (highlighted in orange). These analyses evaluate the performance of the ICRL solver under soft constraints in two distinct aspects:
>
> 1. Aspect One: different reward functions
>
>    In this case, the reward functions differ between the source and target environments. Our results demonstrate that the ICRL solver consistently outperforms the IRC solver, effectively mitigating the impact of variations in reward functions across environments.
>
> 2. Aspect Two: different transition dynamics
>
>    Here, the transition dynamics differ between the source and target environments. The findings indicate that similar to the IRC solver, the ICRL solver can potentially violate constraints when transferring inferred cost functions in soft constraint scenarios due to inferred penalizations compensated by the altered transition dynamics.
>
> ---
>
> References
>
> [1]Vivek S Borkar, “An actor-critic algorithm for constrained markov decision processes,” Systems and control letters, vol. 54, no. 3, pp. 207–213, 2005.
>
> [2] Dotan Di Castro, Aviv Tamar, and Shie Mannor, “Policy gradients with variance related risk
> criteria,” arXiv preprint arXiv:1206.6404, 2012.
>
> [3] Aviv Tamar and Shie Mannor, “Variance adjusted actor critic algorithms,” arXiv preprint
> arXiv:1310.3697, 2013.
>
> [4] Gal Dalal, Krishnamurthy Dvijotham, Matej Vecerik, Todd Hester, Cosmin Paduraru, and Yuval Tassa, “Safe exploration in continuous action spaces,” arXiv preprint arXiv:1801.08757, 2018.
>
> [5] Tsung-Yen Yang, Justinian Rosca, Karthik Narasimhan, and Peter J Ramadge. Projection-based constrained policy optimization. arXiv preprint arXiv:2010.03152, 2020.
>
> [6]  John Schulman, Sergey Levine, Pieter Abbeel, Michael Jordan, and Philipp Moritz. Trust region policy optimization. In International conference on machine learning, pages 1889–1897, 2015.
>
> [7] Liu, Yongshuai, Avishai Halev, and Xin Liu. "Policy learning with constraints in model-free reinforcement learning: A survey." The 30th international joint conference on artificial intelligence (ijcai). 2021.
>
> [8] Yongshuai Liu, Jiaxin Ding, and Xin Liu. Ipo: Interior-point policy optimization under constraints.
> In Proceedings of the AAAI Conference on Artificial Intelligence, volume 34, pages 4940–4947, 2020.
>
> [9] Yinlam Chow, Mohammad Ghavamzadeh, Lucas Janson, and Marco Pavone. Risk-constrained reinforcement learning with percentile risk criteria. The Journal of Machine Learning Research, 18(1):6070–6120, 2017.

---

> > ### Author Response · Authors · 2024-11-29
> > **Author Response to Reviewer MoLg**
> >
> > Dear Reviewer MoLg,
> >
> > We deeply appreciate the invaluable feedback and the thoughtful effort you invested in reviewing our paper! These insightful comments have been invaluable in enhancing both the clarity and the quality of our work.
> >
> > We hope the above response could resolve your concerns. If you have more questions, please feel free to discuss them with us.
> >
> > Thank you very much! Wishing you a joyful Thanksgiving!

---

> > > ### Comment · Reviewer_MoLg · 2024-12-02
> > > **Thanks for your response.**
> > >
> > > Your response's additional experiments and theoretical analysis somewhat relieved my concerns. Synthesize comments from other reviewers, I choose to maintain my score.

---

> > > > ### Author Response · Authors · 2024-12-02
> > > >
> > > > Dear Reviewer MoLg,
> > > >
> > > > We appreciate the significant time and effort dedicated to this process. Thank you very much!
> > > >
> > > > Best regards,
> > > >
> > > > The Authors of Paper 6391

---

### Author Response · Authors · 2024-11-22
**Summary of updates**

We sincerely thank the reviewers (Reviewer MoLg, Reviewer foPH, Reviewer i1Qn, and Reviewer 87Lh) for their insightful and valuable feedbacks, which have been instrumental in improving our work.

We have carefully gone through all comments and incorporated the key updates into the revised manuscript, with changes highlighted in orange for clarity. The updates are listed as follows.

1. We **added experimental results to evaluate soft constraint scenarios** (Remark B.25, Reviewer MoLg).

2. We **included comparisons in more complex environments**, e.g., Half-Cheetah, for IRC and ICRL solvers (Sec 6 and Appendix Sec C, Reviewer i1Qn, 87Lh).

3.  We **improved the clarity and readability of the paper** by clarifying mathematical notations, correcting typos, and adding hyperparameter tables (Sec 4 and Appendix Sec D, Reviewer foPH,  i1Qn).

4. We **provided additional explanations** in the main text when referring to the appendix to enhance clarity (all changes highlighted in orange, Reviewer foPH, 87Lh).

We hope that our revisions can address the concerns raised and look forward to receiving some feedback from the reviewers. We are more than willing to engage in further discussions to refine our work.

---

### Meta-Review · Area_Chair_39Fo · 2024-12-24

**Metareview:**

The paper studies the inverse reward correction (IRC), a recently popularized approach to tackle the problem of inferring safety constraints from expert demonstrations, where the reward that the expert is maximizing is known. The authors show that IRC has better sample complexity but worse generalization across environments with different dynamics or rewards than inverse constrained RL (ICRL), which involves in solving a three-level optimization. Then, they derive conditions in dynamics, rewards, and Lagrangian multipliers that guarantee epsilon-optimality of IRC across different environments.

Here are some positive and negatives from the reviews:
(+) This paper suggests that IRC could be a simpler alternative to ICRL, and thus, can potentially save the practitioners from the complexity of ICRL methods.
(+) The authors have done a good job in motivating the problem and questions posed in the paper.
(-) Moderate to low novelty in the theoretical results as they closely follow prior work on the sample complexity of IRL and transfer of the obtained rewards.
(-) Poor presentation that makes it a challenge to understand all the details. This can explain the low confidence of two reviewers and can be seen from the review of the high-confidence reviewer (Reviewer foPH). Of course, the authors made some improvements in this regard during the rebuttals in response to the reviewers' comments, especially those by Reviewer foPH.

**Additional Comments On Reviewer Discussion:**

The authors addressed some of the reviewers' comments, especially those by Reviewer foPH and improved the clarity of their presentation.

---

### Decision · Program_Chairs · 2025-01-22

Accept (Poster)